

# Estimation of vertical profiles of raindrop size distribution and cloud microphysical processes in stratiform rainfall using vertical-pointing X- and VHF-band radars

Yusuke Goto[1], Taro Shinoda[1], Haruya Minda[1], Moeto Kyushima[1], Hiroyuki Hashiguchi[2], Nozomu
Toda[3], and Shoichi Shige[3]

[1] Institute for Space-Earth Environmental Research, Nagoya University, Nagoya, 4648601, Japan
[2] Research Institute for Sustainable Humanosphere, Kyoto University, Uji, 6110011, Japan
[3] Graduate School of Science, Kyoto University, Kyoto, 6068501, Japan

*Correspondence to*: Yusuke Goto (ygoto@nagoya-u.jp)

**Abstract.**

Simultaneous vertical pointing observations by X- and VHF-band radars were conducted in Japan, and these data were used to estimate vertical profiles of drop size distribution (DSD) parameters of raindrops, assuming a gamma distribution, for a stratiform rainfall event. We used X-band reflectivity, vertical Doppler velocity, and spectral width, combined with VHF-band vertical air motion data. The estimation considers non-Rayleigh scattering and the influence of vertical air motion, and accounts for the contamination of spectral broadening using a forward convolution technique. We demonstrated that for stratiform rainfall, broadening by wind shear may be neglected, even with a relatively coarse radar range resolution of 150 metres. Cloud physical quantities (median volume diameter, liquid water content, normalised intercept parameter) retrieved from the estimated DSD parameters were compared with operational X-band polarimetric radar data and found to be highly accurate. We also point out the potential applicability of this method to satellite-borne radars. Among the estimated parameters, the shape and slope parameters generally increased with decreasing altitude. These changes are attributed to collision-coalescence and breakup based on variations in the cloud physical quantities, likely due to the humid environment. This study suggests that retrieving cloud physical quantities from DSD parameters estimated from vertical observations enables robust discussions on cloud physical processes.

## 1 Introduction

The drop size distribution (DSD) of raindrops is a crucial parameter that is often used to discuss cloud microphysical processes. As DSD is necessary for estimating rain rate and liquid water content ($L_{WC}$ [$gm^{-3}$]; e.g., Fukao and Hamazu, 2014), accurate DSD estimation can lead to a better understanding of precipitation phenomena. The estimation of the rain rate from radar parameters (e.g., radar reflectivity factor ($Z^*$ [$mm^6 m^{-3}$])) is influenced by the variation of the DSD (e.g., Fukao and Hamazu, 2014), leading to potential errors. DSD is also used in numerical models, including cloud physics



parameterisation (e.g., Lin et al., 1983; Murakami, 1990; Ferrier, 1994; Seiki and Nakajima, 2014), and is important for the accurate reproduction and understanding of precipitation. For these reasons, many studies on DSD have been conducted.

Ulbrich (1983) assumed a gamma distribution to show DSD, which is expressed as follows:

$$N(D) = N_0 D^\mu \exp(-\Lambda D), \tag{1}$$

where $D$ [mm] is the equivalent size diameter, $N_0$ [mm$^{-1-\mu}$m$^{-3}$] is the intercept parameter, $\mu$ is the shape parameter, and $\Lambda$ [mm$^{-1}$] is the slope parameter. Tokay and Short (1996) analysed maritime precipitation separately as stratiform and convective rains using ground-based disdrometer data and indicated that stratiform precipitation has more raindrops with larger particle sizes and a smaller number concentration of small particle sizes than convective precipitation at the same precipitation intensity. This means that, assuming a gamma distribution, the value of $\Lambda$ for stratiform precipitation is smaller than that for convective precipitation. Using disdrometer and radar data, Bringi et al. (2003) showed that the location of clustering is different for stratiform and convective precipitation in the relationship between mass-weighted mean diameter ($D_\mathrm{m}$ [mm]) and the normalised intercept parameter ($N_\mathrm{w}$ [mm$^{-1}$m$^{-3}$]). Bringi et al. (2009) and Thompson et al. (2015) also indicated that a line to distinguish between stratiform and convective precipitation can be drawn in the $D_\mathrm{m}$–$N_\mathrm{w}$ plane. Furthermore, Bringi et al. (2003) also noted that, for convective precipitation, the position of clustering in the $D_\mathrm{m}$–$N_\mathrm{w}$ plane is different for maritime and continental precipitation, and that $D_\mathrm{m}$ is smaller and $N_\mathrm{w}$ larger in maritime precipitation than in continental precipitation. Dolan et al. (2018) conducted a principal component analysis using disdrometer observations for various latitude bands and precipitation types and showed that DSDs can be classified into six groups. However, Dolan et al. (2018) also showed that certain groups hardly appear in specific latitudinal zones and that the range of DSD parameters assuming a gamma distribution ($N_0$, $\mu$, and $\Lambda$) varies across latitudinal zones. Thus, DSD varies depending on precipitation type, latitudinal zone, and geography, such as oceanic or continental regions. Recently, studies on DSD using disdrometers have also been increasing in East Asia. For example, Suh et al. (2021) conducted disdrometer observations at multiple sites in South Korea and showed that $D_\mathrm{m}$ increases while $N_\mathrm{w}$ decreases as the distance from the coast increases, even during stratiform precipitation. Seela et al. (2018) examined the DSD of summer and winter rainfall in Taiwan and found that summer rainfall had larger raindrops than winter rainfall. They attributed this to larger ice crystals falling from the upper levels due to the greater convective available potential energy and amount of water vapour in summer. Tsuji et al. (2024) studied oceanic rainfall in Japan using disdrometer data for two years and classified it into two groups with the same precipitation intensity: one with larger raindrop sizes and the other with higher number concentrations. They concluded that the former is characterised by ice-phase processes in deep clouds and the latter by warm rain processes in shallow clouds, based on brightness temperatures observed by satellites. These studies primarily extract parameters related to DSD from ground-based observations and combine them with other data to estimate cloud microphysical processes; however, they do not provide detailed insight into the vertical changes in DSD parameters aloft. As DSD parameters on the ground are the result of various cloud microphysical processes aloft, the simultaneous estimation of cloud microphysical processes and vertical variations in DSD parameters enables a more robust discussion of precipitation formation.



Several methods for estimating the vertical profile of DSD parameters have been proposed. Fukao and Hamazu (2014)
introduced that, assuming a gamma distribution, the DSD can be estimated from $Z^*$, mean vertical Doppler velocity ($\overline{V_\mathrm{d}}$
[m/s]), and Doppler velocity spectrum width ($\sigma_\mathrm{d}$ [m/s]) obtained from a single vertical pointing weather radar if the Rayleigh
scattering approximation is valid and mean vertical air motion ($\overline{V_\mathrm{a}}$ [m/s]) can be ignored. Methods for accounting for the
effect of vertical air motion have also been explored, with several studies highlighting the importance of separating DSD and
$\overline{V_\mathrm{a}}$. VHF- or UHF-band radars are often used to estimate the vertical profile of DSD parameters because the sensitivity of
these radars is influenced by both Bragg scattering from turbulence in clear air and Rayleigh scattering from precipitation
particles (e.g., Wakasugi et al., 1986; Gossard, 1988). However, as the radar cross-section of Rayleigh scattering is inversely
proportional to the fourth power of the wavelength (e.g., Gossard, 1988), the VHF-band radar cannot detect weak rain
(Rajopadhyaya et al., 1993). Similarly, the UHF-band radar also has the disadvantage that Bragg scattering is significantly
overwhelmed by Rayleigh scattering in moderate to heavy rain (Rajopadhyaya et al., 1998). Therefore, instead of a single
profiler method, a dual-frequency method has been proposed that combines VHF- and UHF-band radars to easily separate
the echoes caused by precipitation particles from clear-air turbulence echoes (e.g., Rajopadhyaya et al., 1998; Rao et al.,
2006; Kirankumar et al., 2008). In this method, the spectra obtained from VHF-band radar clearly reflect the influence of
clear-air turbulence, and thus these spectra are applied to the UHF spectra to estimate the rain rate and the median volume
diameter ($D_0$ [mm]). Rajopadhyaya et al. (1998) reported that it is important to correct for the contamination of $\overline{V_\mathrm{d}}$ and $\sigma_\mathrm{d}$
caused by $\overline{V_\mathrm{a}}$ when estimating the rain rate and $D_0$, especially because the uncorrected $\overline{V_\mathrm{d}}$ can be a significant source of error.
The dual-frequency method has also been employed in other frequency band combinations. Pang et al. (2021) combined L-
and C-band vertical pointing radars, used inverse convolution to remove contamination due to turbulence, and then estimated
DSD. Matrosov (2017) indicated the relationship between $D_\mathrm{m}$ and Doppler velocity difference using Ka- and W-band
vertical pointing radars. Williams et al. (2016) calculated the Doppler velocity difference using S- and Ka-band vertical
pointing radars and combined these results with the $Z^*$ and $\sigma_\mathrm{d}$ to estimate the three parameters of the gamma distribution and
$\overline{V_\mathrm{a}}$. As shown above, various frequency bands are used to obtain DSD through vertical observations, and higher-frequency
radars have the benefit of being more sensitive to precipitation particles. Notably, for high-frequency radars, the Rayleigh
scattering approximation becomes invalid for large raindrops (e.g., Ryzhkov and Zrnic, 2019), and the data processing
becomes complex.

Considering both altitude changes in DSD and cloud microphysical processes simultaneously should be useful for
improving cloud physics parameterisations, as altitude variations in DSD are the result of cloud microphysical processes.
Some previous studies have suggested that the predominant cloud microphysical processes, such as collision-coalescence,
collision-breakup, and evaporation, can be inferred from altitude variations in the DSD parameters. For example, Rao et al.
(2006) and Kirankumar et al. (2008) estimated that evaporation is predominant because $\mu$ and/or $\Lambda$ increase with decreasing
altitude in the tropics. Meanwhile, Barthes and Mallet (2013) conducted numerical experiments on a one-dimensional rain
shaft, taking coalescence and breakup into account, and demonstrated that $\mu$ increases as raindrops fall. Williams et al.



(2016) inferred that coalescence is predominant based on a single profile in which $D_m$ increases downward. However, Xie et al. (2016) conducted numerical experiments considering evaporation and demonstrated that $D_0$ can also increase due to evaporation, except for specific DSDs (e.g., initial $\mu < 2$ and $D_0 > 2.3$ mm with 5 mm/h). Thus, even if the altitude

variation of the DSD parameters is the same, the predominant cloud microphysical processes may differ. Such uncertainties in estimating cloud microphysical processes from vertical pointing observations should be avoided by considering the vertical variation of multiple cloud physical quantities (e.g., $D_0$, $L_{WC}$, and $N_w$), which are estimated from retrieved DSD parameters.

Simultaneous vertical observations by X- and VHF-band radars have been conducted in Japan since January 2023. In this

study, we propose methods to investigate the detailed vertical variation of DSD parameters assuming a gamma distribution using X- and VHF-band radar data and to estimate the predominant microphysical processes using cloud physical quantities derived from these estimated DSD parameters. This combination is advantageous for extracting the vertical profiles of DSD parameters because the frequencies are widely separated; they are the frequencies used for rain and vertical air motion observations, respectively. Our method also has the advantage of reducing $\overline{V_a}$ uncertainties by using directly retrieved $\overline{V_a}$ data

from VHF-band radar rather than indirect estimates. We discuss the influence of contamination caused by $\overline{V_a}$, horizontal wind speed, and turbulence on DSD parameter estimation. We consider the non-Rayleigh effect by performing scattering simulations. We also compare the retrieved cloud physical quantities with data from an operational X-band dual-polarimetric radar that performs plan position indicator (PPI) observations and confirm the validity of vertical pointing observations in X- and VHF-band radars. Furthermore, while the next-generation satellite-borne precipitation radar of the Precipitation

Measuring Mission (PMM) is a Ku-band Doppler radar (Nakamura and Furukawa, 2023; Kanemaru et al., 2024), we discuss how accurately DSD can be estimated by using X-band Doppler radars operating at frequencies close to that of Ku-band (e.g., Goto et al., 2025). To rigorously evaluate the accuracy of this estimation method, this study targets stratiform rainfall, which is expected to be less influenced by vertical air motion and non-uniform beam filling than convective rainfall (e.g., Durden et al., 1998). In Section 2, the radars used in this study are introduced. Section 3 presents the methods of data

processing. Section 4 provides an overview of the analysis of a stratiform rainfall case, confirms the accuracy of the DSD estimated from vertical pointing observations by X- and VHF-band radars, and estimates and discusses the cloud microphysical processes that contribute to the change in altitude of the DSD parameters. Section 5 is the summary.

## 2 Instruments

### 2.1 Middle and upper atmosphere (MU) radar

The middle and upper atmosphere (MU) radar in Japan is a Doppler radar with a phased array system, consisting of 475 crossed Yagi antennas, and operates at a frequency of 46.5 MHz, i.e., a VHF-band radar (Fukao et al., 1985; Fukao et al.,



1990). The MU radar is located at the Shigaraki MU Observatory and is 372 m above the mean sea level (AMSL). The specifications of the MU radar are listed in Table 1 and the location is shown in Figure 1. In this study, only vertical pointing

data in tropospheric observation mode, which is characterised by a high range resolution (150 metres), were used to remove the effect of $\overline{V}_a$. As the analysis is limited to the liquid phase, only data at altitudes of about 3400 m or less were used. The mean vertical air motion, as a 1-min value derived by the MU radar ($\overline{V_a^{MU}}$ [m/s]; Toda et al., to be submitted), was used to remove the effects of $\overline{V}_a$ from the vertical Doppler velocity of the X-band radar (NUX, as described in Section 2.2). Notably, $\overline{V_a^{MU}}$ is equivalent to $\overline{V}_a$ in this study. The spectral width of vertical air motion ($\sigma_a$ [m/s]) derived by the MU radar ($\sigma_a^{MU}$

[m/s]; Toda et al., to be submitted) was also used to restrict the time of the analysis (as described in Section 3.2).

Table 1. Specifications of the MU radar in tropospheric observation mode (Fukao et al., 1990; Fukao and Hamazu, 2014).

| Characteristics | Specifications |
| --- | --- |
| Location | Koka City, Shiga Prefecture, Japan (136.106° E, 34.854° N, 372 m AMSL) |
| Type | Monostatic pulse radar (active phased array system) |
| Transmitted frequency | 46.5 MHz |
| Power amplifier | 475 solid state amplifiers |
| Transmitter peak power | 1 MW |
| Pulse repetition frequency | 2,500 Hz |
| Range gate spacing | 150 m |
| Antenna composition | Circular array of 475 crossed Yagi antenna |
| Antenna aperture | 8,330 m$^2$ (103 m in diameter) |
| Beam width | 3.6° |
| Beam directions | Vertical and tilted 10° each to the east, west, north, and south |
| Signal processing technology | Pulse compression |
| Doppler processing | FFT |
| Minimum observation height (AMSL) | 1,422 m (vertical pointing) |



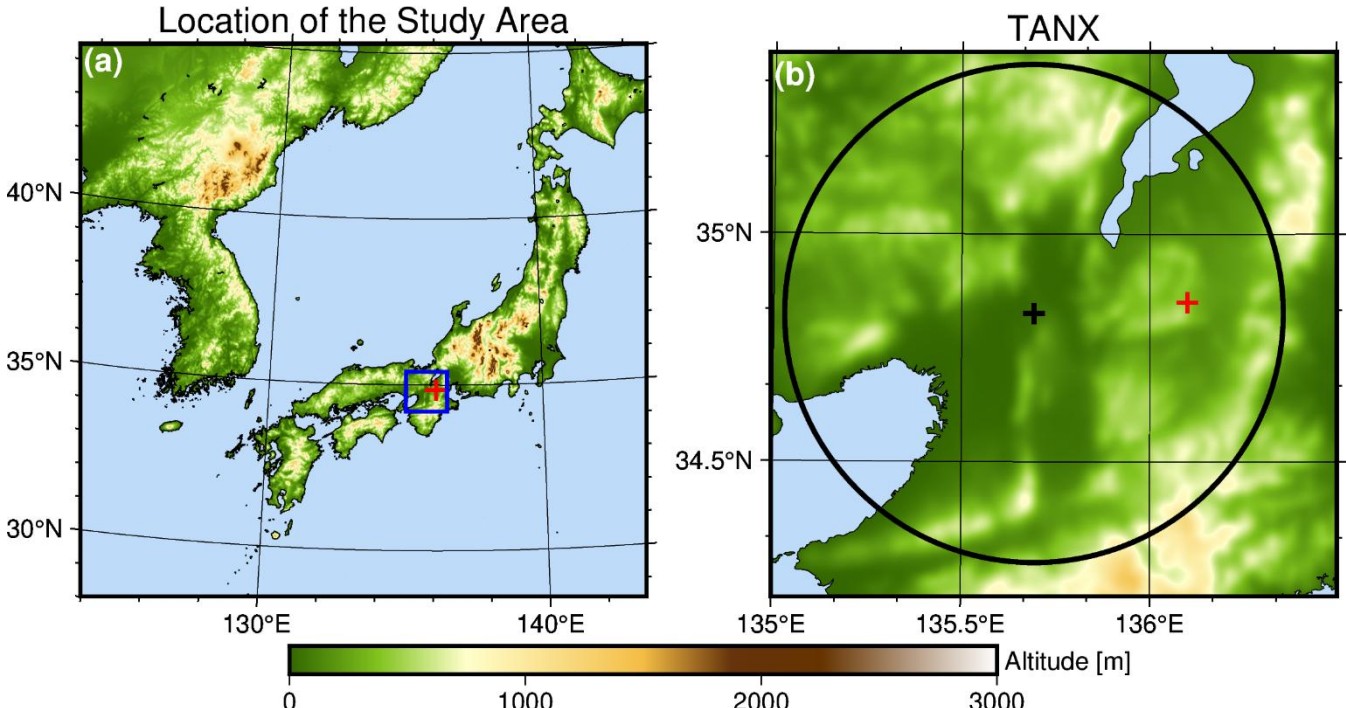

Figure 1. (a) The location of the study area. The blue rectangle shows the areas in (b). The red + marks in (a) and (b) indicate the location of the Shigaraki MU Observatory. The black + mark and circle in (b) indicate the location of the operational X-band polarimetric radar (Tanokuchi radar; TANX) that was used for validation and its observation range (radius of 60 km), respectively.

## 2.2 X-band radar of Nagoya University (NUX)

The X-band radar of Nagoya University (Morotomi et al., 2012; hereafter NUX) is a mobile polarimetric Doppler radar. The specifications are presented in Table 2. The NUX has been installed within the Shigaraki MU Observatory (Figure 1) since January 2023 and conducts vertical pointing observations for 4 min of every 5 min. During vertical observations, the NUX rotates 360 degrees in azimuth every 30 s. The radar reflectivity factor ($Z^{*\mathrm{NUX}}$ [mm$^6$m$^{-3}$]), mean vertical Doppler velocity of the spectrum ($\overline{V_\mathrm{d}^{\mathrm{NUX}}}$ [m/s]), and Doppler velocity spectral width ($\sigma_\mathrm{d}^{\mathrm{NUX}}$ [m/s]) obtained from the NUX were used to estimate the vertical profile of DSD parameters assuming a gamma distribution. As NUX does not output raw spectral information, the Doppler velocity spectral width, processed internally and output by the radar, was used in this study. The $Z^*$ is often expressed as the radar reflectivity ($Z = 10 \log_{10} Z^*$ [dBZ]), and the $Z$ obtained from the NUX ($Z^{\mathrm{NUX}}$ [dBZ]) was also used.

Table 2. The specifications of the NUX during vertical observation and at an elevation angle of 20°.

| Characteristics | Specifications |
| --- | --- |





| Location | Koka City, Shiga Prefecture, Japan (136.109° E, 34.852° N, 384 m AMSL) |
|---|---|
| Type | Dual-polarisation Doppler pulse radar |
| Transmitted frequency | 9.375 GHz |
| Power amplifier | Solid state amplifier |
| Peak power | 200 W |
| Pulse repetition frequency | 1,600 Hz and 2,000 Hz (Dual PRF) |
| Nyquist velocity | 64 m/s (12.8 m/s and 16.0 m/s) |
| Max range | 64 km |
| Pulse width | Short pulse: 1 µs |
| | Long pulse: 32 µs |
| Range gate spacing | 150 m |
| Antenna type | Parabola |
| Antenna diameter | 2.0 m |
| Beam width | 1.1° |
| Rotation rate | 2 rpm (vertical observation) / 1 rpm (elevation angle of 20°) |
| Signal processing technology | Pulse compression |
| Doppler processing | Pulse-pair |
| Minimum observation height (AMSL) | 1584 m |

We also used $\overline{V_d^{NUX}}$ data at an elevation angle of 20° for the calibration of $\sigma_d^{NUX}$. Observations at an elevation angle of 20° were conducted once every five minutes.

## 2.3 Operational X-band polarimetric radar (TANX)

For comparison of estimated cloud physical quantities and dominant cloud microphysics, data from an operational X-band polarimetric Doppler radar located about 38 km away from the Shigaraki MU Observatory were also used (Figure 1b). This X-band radar is one of the radars in the eXtended RAdar Information Network (XRAIN) operated by the Ministry of Land, Infrastructure, Transport and Tourism in Japan, and this study utilises the radar at Tanokuchi station (TANX). The specifications are presented in Table 3. The TANX performs one volume scan at 12 elevation angles every 5 min, while conducting multiple low-elevation scans, resulting in a total of 15 PPI scans every 5 min. The elevations of the TANX are set between 0.9 and 15.0 degrees. The variables used from the TANX were the horizontal radar reflectivity factor ($Z^{*TANX}$ [mm⁶/m³]), horizontal radar reflectivity ($Z^{TANX}$ [dBZ]), differential reflectivity ($Z_{DR}^{TANX}$ [dB]), and specific differential phase ($K_{DP}^{TANX}$ [°/km]).



Table 3. Specifications of the TANX.

| Characteristics | Specifications |
|---|---|
| Location | Hirakata City, Osaka Prefecture, Japan (135.692° E, 34.826° N, 96 m AMSL) |
| Type | Dual-polarisation Doppler pulse radar |
| Transmitted frequency | 9.725 GHz |
| Power amplifier | Klystron |
| Peak power | 100 kW |
| Pulse repetition frequency | 1,440 Hz and 1,800 Hz (Dual PRF) |
| Max range | 60 km |
| Pulse width | 1 μs |
| Range gate spacing | 150 m |
| Antenna type | Parabola |
| Antenna diameter | 2.2 m |
| Beam width | 1.2° |
| Rotation rate | 3.5 rpm or 4.5 rpm |
| Signal processing technology | Non-pulse compression |

## 3 Data processing

### 3.1 NUX variables used for estimating DSD parameters

The reflectivity factor ($Z^*$), mean vertical Doppler velocity ($\overline{V_\mathrm{d}}$), and Doppler velocity spectral width ($\sigma_\mathrm{d}$) assuming Rayleigh scattering approximation, obtained from the vertical pointing radar observations, are expressed by the following equations, respectively (Fukao and Hamazu, 2014):

$$Z^* = \int_{D_\mathrm{min}}^{D_\mathrm{max}} D^6 N(D)\,\mathrm{d}D \,, \tag{2}$$

$$\overline{V_\mathrm{d}} = -\frac{\int_{D_\mathrm{min}}^{D_\mathrm{max}} V_\mathrm{t}(D) D^6 N(D)\,\mathrm{d}D}{\int_{D_\mathrm{min}}^{D_\mathrm{max}} D^6 N(D)\,\mathrm{d}D} + \overline{V_\mathrm{a}}, \tag{3}$$

$$\sigma_\mathrm{d} = \sqrt{\frac{\int_{D_\mathrm{min}}^{D_\mathrm{max}}\{V_\mathrm{t}(D) + \overline{V_\mathrm{a}}\}^2\, D^6 N(D)\,\mathrm{d}D}{\int_{D_\mathrm{min}}^{D_\mathrm{max}} D^6 N(D)\,\mathrm{d}D} - \left(\frac{\int_{D_\mathrm{min}}^{D_\mathrm{max}} V_\mathrm{t}(D)\, D^6 N(D)\,\mathrm{d}D}{\int_{D_\mathrm{min}}^{D_\mathrm{max}} D^6 N(D)\,\mathrm{d}D} + \overline{V_\mathrm{a}}\right)^2}\,, \tag{4}$$



where $V_t(D)$ [m/s] is the fall velocity of the precipitation particles, and the $D_{max}$ and $D_{min}$ [mm] are the maximum and minimum sizes of the precipitation particles, respectively. As this study deals with raindrops, the $V_t(D)$ was used with the following equation (Atlas et al., 1973; Fukao and Hamazu, 2014),

$$V_t(D) = (9.65 - 10.3e^{-0.6D}) \left(\frac{\rho_{a0}}{\rho_a}\right)^{0.4}, \tag{5}$$

where the $\rho_{a0}$ and $\rho_a$ [kg/m³] are the air density of the International Standard Atmosphere (= 1.225 kg/m³) and the ambient air, respectively. The $\rho_a$ is calculated by linear interpolation of pressure and temperature data from ERA5 (Hersbach et al., 2020) to the observed grid points and using the equation of state for a dry atmosphere. As ERA5 collects hourly data, the data from ERA5 closest to the vertical observation time is used. For Eqs. 3 and 5, the fall velocity of a raindrop is generally expressed as a positive value, while the Doppler velocity and vertical air motion are expressed as a positive value in the

direction away from the ground-based radar. To adjust the sign, the first term on the right side of Eq. 3 is multiplied by $-1$.

Equations 2–4 assume that the Rayleigh scattering approximation holds; however, the X-band radar (NUX) could produce errors resulting from the limitations of the Rayleigh scattering approximation. Figure 2 shows the relationship between raindrop diameter and backscattering cross section of the NUX, calculated by scattering simulations using the T-matrix method (Waterman, 1971) with the PyTmatrix package (Leinonen, 2014), for raindrop diameters up to a maximum of 9 mm

(Pruppacher and Klett, 1997). The figure indicates that when the diameter exceeds about 2.5 mm, the resonance in the induced electric field becomes significant and the Rayleigh scattering approximation is no longer valid. Therefore, in this study, non-Rayleigh scattering was also considered following Williams et al. (2016), and is expressed as follows:

$$Z^{*nonRay} = \frac{\lambda^4}{\pi^5 |K|^2} \int_{D_{min}}^{D_{max}} \sigma_b(D)N(D)dD, \tag{6}$$

$$\overline{V_d^{nonRay}} = -\frac{\int_{D_{min}}^{D_{max}} V_t(D)\sigma_b(D)N(D)dD}{\int_{D_{min}}^{D_{max}} \sigma_b(D)N(D)dD} + \overline{V_a}, \tag{7}$$

$$\sigma_d^{nonRay} = \sqrt{\frac{\int_{D_{min}}^{D_{max}} \{V_t(D) + \overline{V_a}\}^2 \sigma_b(D)N(D)dD}{\int_{D_{min}}^{D_{max}} \sigma_b(D)N(D)dD} - \left(\frac{\int_{D_{min}}^{D_{max}} V_t(D)\sigma_b(D)N(D)dD}{\int_{D_{min}}^{D_{max}} \sigma_b(D)N(D)dD} + \overline{V_a}\right)^2}, \tag{8}$$

where $\lambda$ [mm] is the wavelength of the radar, $|K|^2$ is the dielectric factor with a value of 0.93 for raindrops and X-band radars (Gunn and East, 1954), and $\sigma_b$ [mm²] is the backscattering cross section. The first term on the right side of Eq. 7 is equivalent to $\overline{V_{tz}^{nonRay}}$ [m/s].




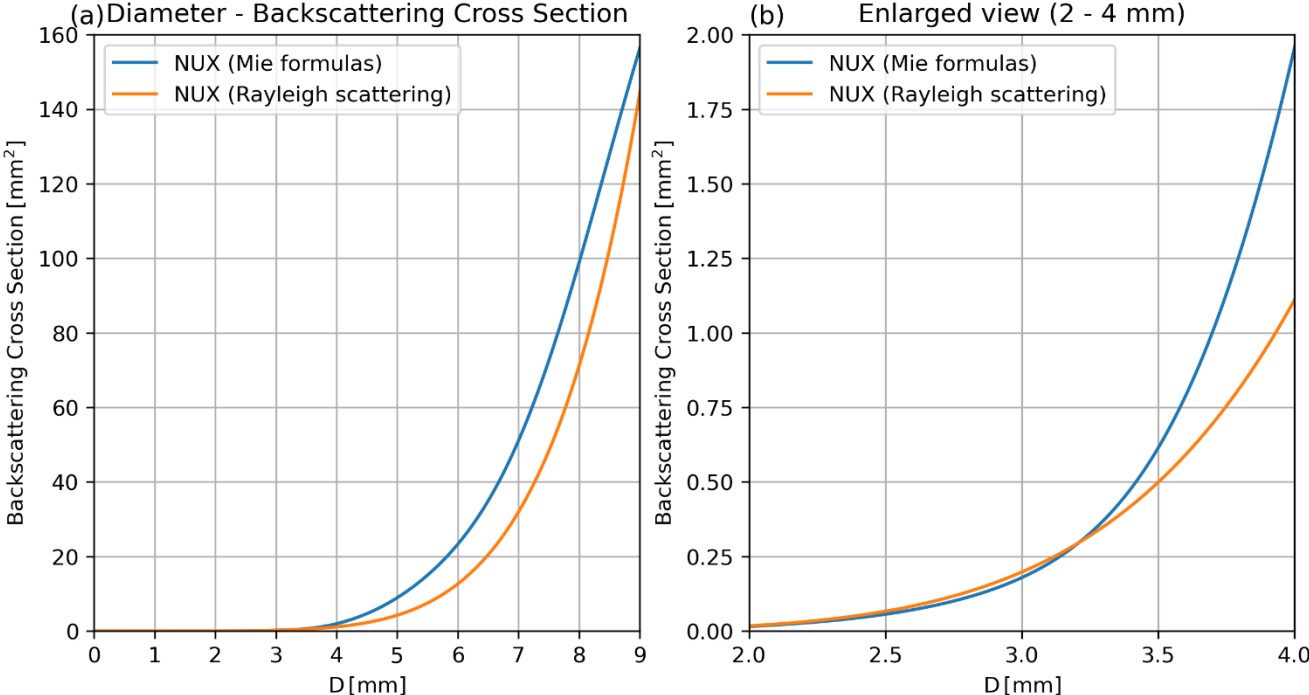

Figure 2. (a) Relationship between the size of a sphere and the backscattering cross section, and (b) an enlarged view. The blue line is obtained from a scattering simulation using the T-matrix method, and the orange line assumes Rayleigh scattering. For this simulation, the wavelength is 32 mm (X-band), the raindrop temperature is 10 °C, and the raindrop shape is assumed to be a sphere.

## 3.2 Reducing the influence of $\sigma_\mathrm{d}^\mathrm{NUX}$ broadening

The $\sigma_\mathrm{d}$ from vertical pointing observations is mainly due to the size spread of precipitation particles ($\sigma_\mathrm{DSD}$); however, the spread is also influenced by horizontal wind, turbulence, and wind shear (e.g., Shupe et al., 2008; Williams et al., 2016). Therefore, $\sigma_\mathrm{d}^\mathrm{NUX}$ is slightly larger owing to various contaminations, although the estimation of DSD should only consider the $\sigma_\mathrm{DSD}$. This broadening is expressed as the sum of the variances, as follows (Shupe et al., 2008; Williams et al., 2016):

$$\sigma_\mathrm{broadening}^2 = \sigma_\mathrm{H}^2 + \sigma_\mathrm{T}^2 + \sigma_\mathrm{S}^2, \tag{9}$$

where $\sigma_\mathrm{H}^2$, $\sigma_\mathrm{T}^2$, and $\sigma_\mathrm{S}^2$ are terms attributed to horizontal wind speed, turbulence, and wind shear, respectively. Williams et al. (2016) pointed out that $\sigma_\mathrm{H}^2$ and $\sigma_\mathrm{T}^2$ are particularly significant and also noted that corrections should be applied for radar beamwidths of 3° or greater. Rajopadhyaya et al. (1998) indicated that the estimated $D_\mathrm{m}$ would have an error of about 15% if no correction was made for the $\sigma_\mathrm{d}$ broadening of the UHF-band radar. Notably, the UHF-band radar beamwidth used in Rajopadhyaya et al. (1998) is 9°. The beam width of the NUX used in this study is 1.1° (Table 2) and the contamination due to broadening is expected to be small; thus, efforts should be made to minimise errors because the influence of broadening



varies depending on the surrounding atmospheric conditions, even when the beamwidth is relatively small. Gossard and Strauch (1983) expressed the broadening variance due to horizontal wind ($\sigma_H^2$) using the following equation:

$$\sigma_H^2 = \frac{U^2 \varphi^2}{2.76}, \tag{10}$$

where $U$ [m/s] is horizontal wind speed and $\varphi$ [radian] is radar beam width. Furthermore, assuming an inertial sub-range, the contribution of turbulence is given by the following equation (Shupe et al., 2008):

$$\frac{\sigma_T^2}{\sigma_{measurement}^2} \approx \frac{L_s^{2/3}}{L_l^{2/3} - L_s^{2/3}}, \tag{11}$$

where $\sigma_{measurement}^2$ is the Doppler velocity width obtained from observations, and in this study, it is the variance of $\sigma_d^{NUX}$ over 30-second interval. In this study, $L_s$ is the length of the scattering volume for 0.1-s dwell time and $L_l$ indicates the larger

eddies passing through the effective sample volume that results from an average of the radar observations over 30-s in this study. $L$ is expressed by the following equation (Shupe et al., 2008):

$$L = Ut + 2R \sin\left(\frac{\varphi}{2}\right), \tag{12}$$

where $t$ [s] is time and $R$ [m] is the distance from the radar to the observation grid point. Equations 10 and 12 show that the horizontal wind aloft is important in addition to the radar beam width. Additionally, the effect of horizontal and vertical

shear in the vertical winds is expressed by the following equation (Gossard and Strauch, 1983; Shupe et al., 2008):

$$\sigma_S^2 = \frac{k_h^2 R^2 \varphi^2}{2.76} + \frac{k_v^2 \Delta^2}{12}, \tag{13}$$

where $k_h$ and $k_v$ [s$^{-1}$] are the horizontal shear in vertical wind and vertical shear in vertical wind, respectively. $\Delta$ [m] is the range gate spacing. Note that while the effects of wind shear are often neglected in previous studies (e.g., Williams et al., 2016; Pang et al., 2021), the range resolution of the NUX used in this study is 150 metres (Table 2), which is about 2 to 5

times coarser than that of previous studies. Therefore, while the effects of shear should be evaluated in this study, efforts to reduce broadening are also necessary. Rao et al. (1999) gave an example of the vertical structure of $\overline{V_a}$ and $\sigma_a$ in stratiform and convective clouds based on VHF-band radar data, showing that the absolute value of $\overline{V_a}$ and the value of $\sigma_a$ are both smaller in stratiform clouds than in convective clouds. Therefore, in this study, the method for estimating the vertical profile of DSD parameters was applied only to typical stratiform rainfall, which is expected to have small temporal and vertical

wind variations. Based on the data presented by Rao et al. (1999), typical stratiform rainfall is defined as rainfall that satisfies the following conditions: (I) $\left|\overline{V_a^{MU}}\right| \leq 1.0$ m/s for the entire liquid phase and (II) a mean of $\sigma_a^{MU}$, denoted as $(\overline{\sigma_a^{MU}})$, is $\leq 1.2$ m/s for the liquid phase. Notably, the beamwidth of the VHF-band radar used by Rao et al. (1999) was 3.0°, and that of the MU radar is 3.6° (Table 1); thus, small differences in $\sigma_a$ due to beamwidth are expected. The 0 °C altitude was identified using ERA5, the thickness of the melting layer was set as 1500 m following Rosenfeld et al. (1995), and the zone

below the melting layer was defined as the liquid phase.





To consider the influence of $\sigma_d^{NUX}$ broadening, the wind speed aloft must be estimated. We applied the Velocity Azimuth Display (VAD) method (Browning and Wexler, 1968) using Doppler velocity data of NUX ($\overline{V_d^{NUX}}$) at an elevation angle of 20° to estimate the wind speed aloft. In this study, calculations were performed using the linear least squares method (Tsuboki and Wakahama, 1988). Figure 3 shows a typical example of the horizontal wind speed aloft estimated using the

VAD method based on the Doppler velocity from NUX. Throughout the analysis period, a west-southwesterly wind (as shown in Figure 3b) prevailed for much of the time, with speeds of 10 m/s to 30 m/s (as shown in Figure 3a).  For such wind fields, $\sigma_H^2$ (Eq. 10) is on the order of $10^{-2}$ to $10^{-1}$. Additionally, we have confirmed that the $\sigma_{measurement}^2$ of NUX is around 0.03 m²/s² (not shown), and that $\sigma_T^2$ (Eq. 11) ranges from $10^{-3}$ to $10^{-2}$ in magnitude. This implies that $\sigma_H^2$ and $\sigma_T^2$ may affect the estimation of DSD parameters; therefore, this study corrects for the broadening caused by these factors.


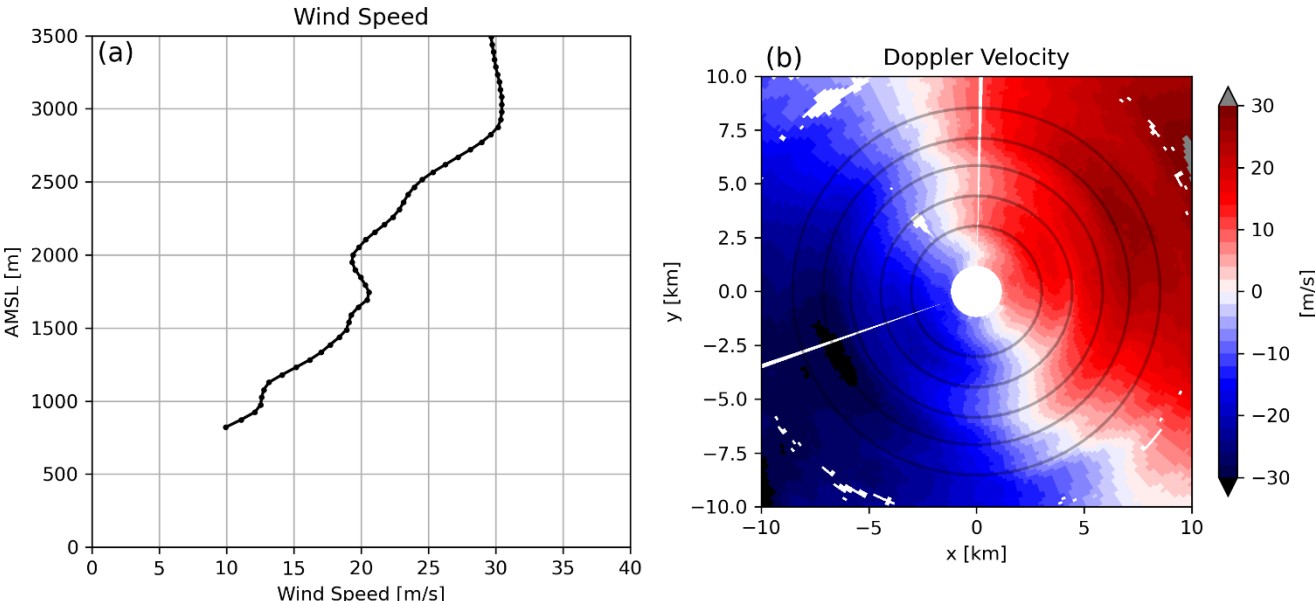

Figure 3. (a) Horizontal wind speed estimated using the VAD method and (b) horizontal distribution of NUX Doppler velocity at 15:39 JST on 2 June 2023. In (b), NUX is at the centre; warm colours indicate Doppler velocity away from the radar, and cool colours indicate Doppler velocity toward the radar. The grey circles in (b) indicate, from the innermost to the outermost, locations at 1500 m, 2000 m, 2500
m, 3000 m, and 3500 m altitude.

Regarding corrections for shear, we focus on the vertical shear of vertical winds. Figure 4 shows the vertical change in vertical air motion derived from the $\overline{V_a^{MU}}$, and even at the 99th percentile, $k_v$ is calculated to be 0.0023 s⁻¹. Even when this 99th percentile value is substituted into the second term on the right-hand side of Eq. 13, the estimated variance remains on

the order of $10^{-3}$, even with Δ= 150 metres. This indicates that the effect of $\sigma_S^2$ is negligible because the values are less than



0.01 m²/s² in most cases (e.g., Pang et al., 2021), during the defined period of typical stratiform rainfall, even for radars with a relatively coarse range gate length (~ 150 metres). Note that the beam width at a distance of 3000 m from NUX is approximately 60 m, which is less than half the range gate length. Therefore, the broadening effect due to horizontal shear in vertical air motions is expected to be smaller than that due to vertical shear in vertical air motions, and can thus be neglected.


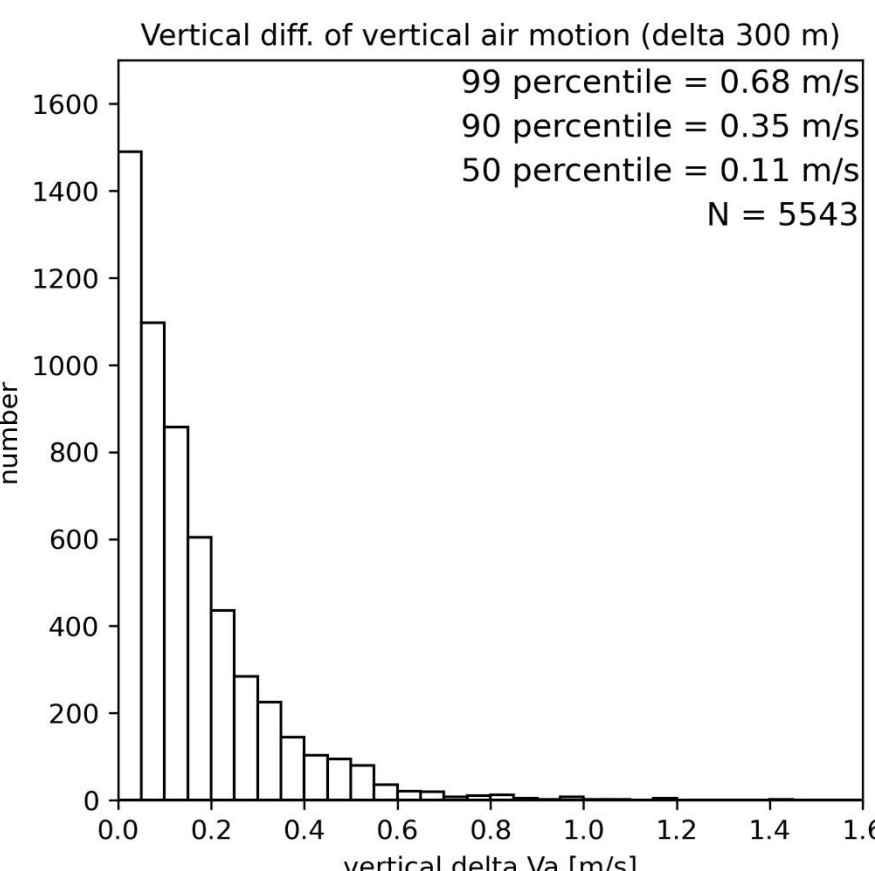

Figure 4. Histogram of vertical changes in vertical air motion during typical stratiform rainfall, determined from MU radar data. N is the total number of this samples.

**3.3 Forward calculation of the Doppler spectrum via convolution of DSD effects and broadening**

Several previous studies (e.g., Shupe et al., 2008; Williams et al., 2016) have assumed the following equation and removed the broadening effect through simple subtraction from $\sigma_d^2$:

$$\sigma_d^2 \approx \sigma_{DSD}^2 + \sigma_{broadening}^2. \tag{14}$$



However, this assumption applies only when the radar reflectivity-weighted Doppler spectral density at each Doppler
velocity ($v$) [m/s] generated by DSD in still air ($S_{\text{DSD}}(v)$) [(mm$^6 \cdot$ m$^{-3}$)$\cdot$(m$\cdot$s$^{-1}$)$^{-1}$] can be assumed to be Gaussian (Williams et
al., 2016). As the DSD does not necessarily follow a Gaussian distribution (e.g., black line in Figure 5a), the convolution
calculation should be considered rigorously. Therefore, we use the following convolution calculations (e.g., Zhu et al., 2023)
for a rigorous estimation of the DSD parameters:

$$S_{\text{Doppler}}(v) = S_{\text{DSD}}(v) \otimes S_{\text{broadening}}(v), \tag{15}$$

where $S_{\text{Doppler}}(v)$ [(mm$^6 \cdot$ m$^{-3}$) $\cdot$ (m $\cdot$ s$^{-1}$)$^{-1}$] is the radar reflectivity-weighted Doppler velocity spectral density and
$S_{\text{broadening}}(v)$ [(m $\cdot$ s$^{-1}$)$^{-1}$] is the spectral broadening due to the horizontal wind speed and turbulence. The $\otimes$ symbol
represents convolution. Note that $S_{\text{DSD}}(v)$ and $S_{\text{broadening}}(v)$ are expressed by the following equations, respectively (Zhu et
al., 2023):

$$S_{\text{DSD}}(v) = N(D)\sigma_{\text{b}}(D)\frac{\mathrm{d}D}{\mathrm{d}V_{\text{t}}}, \tag{16}$$

$$S_{\text{broadening}}(v) = \frac{1}{\sigma_{\text{broadening}}\sqrt{2\pi}} \times \exp\left\{-\frac{1}{2}\left(\frac{v}{\sigma_{\text{broadening}}}\right)^2\right\}. \tag{17}$$

Figure 5 shows examples of the shape variations in the normalised Doppler spectral density when varying the values of $\mu$
and $\Lambda$ at $D_0 = 1.1$ mm. When $\mu = -1$ (Figure 5a), the shape is less smooth than that in Figure 5b due to the influence of
non-Rayleigh scattering. The influence of broadening is reflected through the convolution calculation. The observed $\sigma_{\text{d}}^{\text{NUX}}$
corresponds to the theoretical standard deviation calculated after accounting for the broadening shown in Figure 5.





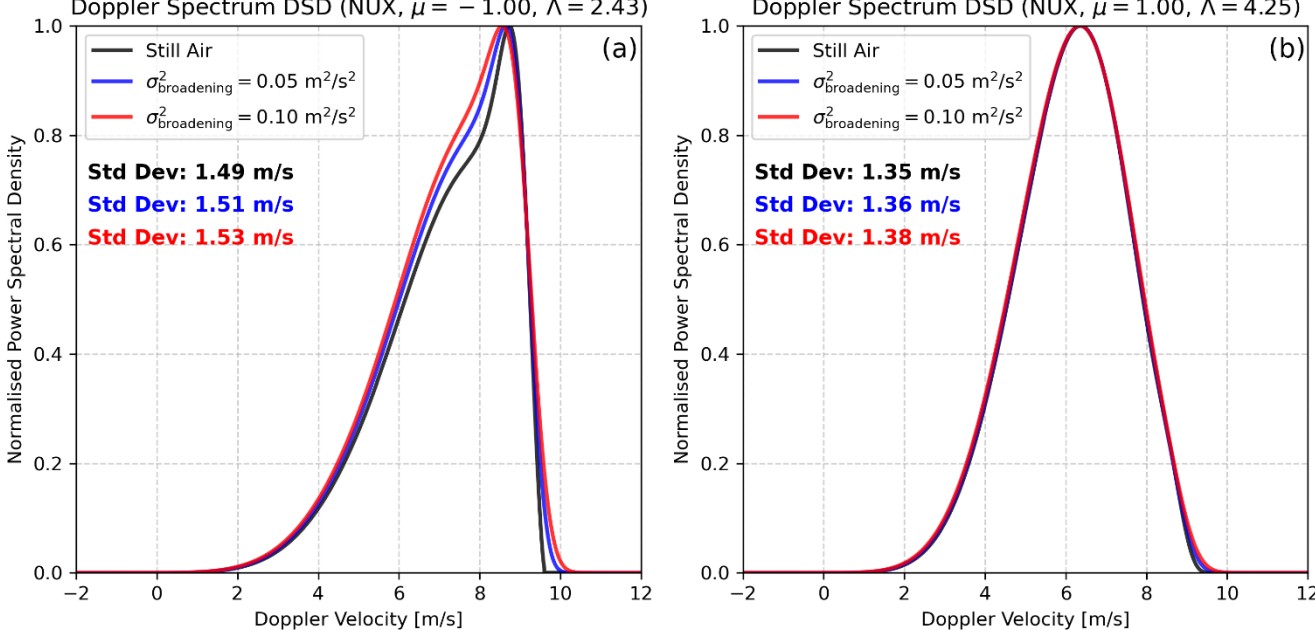

Figure 5. Normalised Doppler spectral density calculated for a gamma DSD with $D_0 = 1.1$ mm, when (a) $\mu = -1$ and (b) $\mu = 1$ are applied. The wavelength is assumed to be the same as that of NUX (= 32 mm), and the effects of non-Rayleigh scattering are also considered using results from scattering simulations performed with the T-matrix method. The black line represents still air conditions, while the blue and red lines indicate broadening variance of 0.05 m²/s² and 0.10 m²/s², respectively. The Std Dev in the figure represents the standard deviation of the spectrum, with the colour corresponding to the $\sigma^2_{\text{broadening}}$ magnitudes or still air.

### 3.4 Methods for estimating the vertical profile of DSD parameters

The theoretical values of $\overline{V_{\text{tz}}^{\text{nonRay}}}$ $\left(\overline{V_{\text{tz}}^{\text{th}}}\right)$ and $\sigma_{\text{d}}^{\text{nonRay}}$ $\left(\sigma_{\text{d}}^{\text{th}}\right)$ with respect to the values of $\mu$, $\Lambda$, and $\sigma_{\text{broadening}}$ are obtained as shown in Figure 6. Notably, $N_0$ has to be assumed in the scattering simulation; however, the $\overline{V_{\text{tz}}^{\text{nonRay}}}$ and $\sigma_{\text{d}}^{\text{nonRay}}$ are independent of $N_0$ (e.g., Fukao and Hamazu, 2014), hence $N_0$ is set to 8000 $\text{mm}^{-1-\mu}\text{m}^{-3}$ following Marshall and Palmer (1948) in Figure 6. In Figure 6, the wavelength of the radar is set to match the NUX ($\approx 32$ mm) and the raindrop temperature is assumed to be 10 °C. The raindrop shape is based on Thurai et al. (2007), $D_{\text{max}}$ is set to 9 mm (Pruppacher and Klett, 1997), $D_{\text{min}}$ is set to 0.01 mm, the interval of $D$ is divided and integrated into 1024 segments, and the $\mu$, $\Lambda$, and $\sigma_{\text{broadening}}$ intervals are all 0.01 in the scattering simulation. Convolving the broadening effect increases the $\sigma_{\text{d}}^{\text{th}}$ value, with a particularly pronounced increase in regions where $\Lambda$ is small (Figure 6c). Note that as $\sigma^2_{\text{broadening}}$ equals 0.10 m²/s², $\sigma_{\text{broadening}}$ equals 0.32 m/s (Figure 6b). $\overline{V_{\text{tz}}^{\text{th}}}$ is unaffected by broadening and therefore is independent of the value of $\sigma_{\text{broadening}}$ (Figure 6d).





Figure 6. Theoretical values of (a) $\sigma_d^{th}$ in still air, (b) $\sigma_d^{th}$ when the broadening has a variance of 0.10 m²/s², (c) difference in $\sigma_d^{th}$ ((b) - (a)), and (d) $\overline{V_{tz}^{th}}$ with respect to combinations of $\mu$ and $\Lambda$ obtained from scattering simulations using the T-matrix method. For this simulation, the wavelength is 32 mm, the raindrop temperature is 10 ℃, the raindrop shape follows Thurai et al. (2007), $D_{max}$ is 9 mm, $D_{min}$ is 0.01 mm, and the calculation intervals for $\mu$, $\Lambda$, and $\sigma_{broadening}$ are set to 0.01. $N_0$ is set to 8000 mm$^{-1-\mu}$m$^{-3}$. The calculation ranges for $\mu$ and $\Lambda$ are as shown in this figure, and $\sigma_{broadening}$ is calculated from 0.00 (still air) to 1.19 m/s.





The procedure for estimating DSD parameters is described. The horizontal wind speed at altitude estimated using the VAD method is linearly interpolated to match the grid point heights of the NUX vertical observations, deriving the horizontal wind speed at each grid point. Additionally, we obtain the variance of $\sigma_d^{\text{NUX}}$ in vertical observations every 30

seconds for each altitude. Subsequently, $\sigma_{\text{broadening}}^2$ is obtained using Eqs. 9–12 and converted to standard deviation. This process yields the optimal $\sigma_{\text{broadening}}$ for each time and altitude, enabling the search for DSD parameters thereafter.

For $Z^{*\text{nonRay}}$, $\overline{V_d^{\text{nonRay}}}$, $\overline{V_{tz}^{\text{nonRay}}}$, $\sigma_d^{\text{nonRay}}$, and $\overline{V_a}$ in Eqs. 6–8, the variables obtained from the observations in this study are expressed as $Z^{*\text{NUX}}$, $\overline{V_d^{\text{NUX}}}$, $\overline{V_{tz}^{\text{NUX-MU}}}$ (or $\overline{V_{tz}^{\text{NUX}}}$), $\sigma_d^{\text{NUX}}$, and $\overline{V_a^{\text{MU}}}$, respectively. First, one NUX profile is obtained as a 30-s value while the MU radar profile is derived as a 1-min value; thus, this study assumed that $\overline{V_a}$ remained the same for 1

min and the vertical profile of DSD parameters was estimated every 30 s. Second, the MU radar data are linearly interpolated to match the NUX grid points due to the difference in installation altitudes between the MU radar and the NUX (Table 1 and Table 2). Third, Eq. 7 is transformed into the following equation to obtain $\overline{V_{tz}^{\text{NUX-MU}}}$,

$$\overline{V_{tz}^{\text{NUX-MU}}} = \overline{V_d^{\text{NUX}}} - \overline{V_a^{\text{MU}}}. \tag{18}$$

Next, the optimal combination of $\mu$ and $\Lambda$ that minimises the error is obtained using the least-squares method expressed in

the following equation,

$$O_{pt}\left(\mu, \Lambda, \sigma_{\text{broadening}}\right) = \left\{\overline{V_{tz}^{\text{NUX-MU}}} - \overline{V_{tz}^{\text{th}}}(\mu, \Lambda)\right\}^2 + \left\{\sigma_d^{\text{NUX}} - \sigma_d^{\text{th}}\left(\mu, \Lambda, \sigma_{\text{broadening}}\right)\right\}^2. \tag{19}$$

Thus, the optimal $\mu$ and $\Lambda$ are estimated by comparing the observation with theoretical values to which the estimated broadening has been added (i.e., a forward modelling approach). Finally, combining Eqs. 1 and 6, $N_0$ is calculated using the optimal combination of $\mu$ and $\Lambda$ and the following equation,


$$N_0 = \frac{Z^* \pi^5 |K|^2}{\lambda^4 \int_{D_{\min}}^{D_{\max}} \sigma_b(D) D^\mu e^{-\Lambda D} dD}. \tag{20}$$

Notably, only in Section 4.5, to confirm the importance of considering the effect of $\overline{V_a}$, similar to Rajopadhyaya et al. (1998), we also estimated the DSD parameters assuming $\overline{V_{tz}^{\text{NUX}}} = \overline{V_d^{\text{NUX}}}$.

Figure 6 shows a range of $-7 \leq \mu \leq 25$, whereas fitting a gamma distribution often imposes a limit on the range of $\mu$ values (e.g., Kirankumar et al., 2008; Dolan et al., 2018). In this study, only grid points for which the optimal values of $\mu$

derived from Eq. 19 were within the range of $-3 \leq \mu \leq 15$ were used in the analysis. This screening process removed about 1.4% of the total data.

## 3.5 Methods for comparing the cloud physical quantities

DSD parameters ($N_0$, $\mu$, and $\Lambda$) are difficult to estimate directly and exactly from polarimetric parameters, especially in

radars operating at high scanning speeds (Keat et al., 2016). Therefore, the cloud physical quantities ($D_0$, $L_{\text{WC}}$, and $N_w$) aloft





retrieved from the NUX and MU radar-based estimated DSD parameters were compared with the TANX data to confirm the validity of the vertical pointing observations in this study. Several methods have been proposed to estimate the $D_0$ and $L_{WC}$ aloft using polarimetric parameters obtained from X-band dual polarisation radars. For the estimation of $D_0$, Anagnostou et al. (2008), Kim et al. (2010), and Matrosov et al. (2005) expressed the following equations, respectively:

$$D_0 = 0.5 + 1.5Z_{DR} - 0.4Z_{DR}^2 + 0.03Z_{DR}^3, \tag{21}$$

$$D_0 = 0.79 + 0.77Z_{DR}, \tag{22}$$

$$D_0 = 1.46Z_{DR}^{0.49}. \tag{23}$$

For the estimation of $L_{WC}$, Anagnostou et al. (2008) expressed the following equation:

$$L_{WC} = 10^{-2.9}(Z^*)^2 10^{-2.48Z_{DR}+1.72Z_{DR}^2-0.5Z_{DR}^3+0.06Z_{DR}^4}. \tag{24}$$

Maki et al. (2005) also proposed equations to estimate $L_{WC}$ using DSD derived from disdrometer data observed in Japan, which was expressed as:

$$L_{WC} = 0.00393(Z^*)^{0.55}, \tag{25}$$

$$L_{WC} = 0.991K_{DP}^{0.713}. \tag{26}$$

Although $K_{DP}$ is considered less sensitive and unreliable for small raindrops with small oblateness, Godo et al. (2014) found

that $K_{DP}$ *the* of XRAIN is sensitive enough at 0.1 °/km or more. The $K_{DP}$ used in the following analysis was confirmed to be greater than 0.1 °/km, therefore, no sensitivity issues are expected. The $N_w$ is expressed using the $D_0$ and $L_{WC}$ (e.g., Bringi and Chandrasekar, 2001) as follows:

$$N_w = \frac{3.67^4}{\pi \rho_w} \left( \frac{1000 L_{WC}}{D_0^4} \right), \tag{27}$$

where $\rho_w$ [g/cm³] is the density of liquid water (1 g/cm³).

The estimated DSD parameters, assuming a gamma distribution from vertical pointing observations, can also be used to obtain cloud physical quantities. The $L_{WC}$ is defined by the following equation (e.g., Fukao and Hamazu, 2014; Williams et al., 2016):

$$L_{WC} = \frac{0.001\pi\rho_w}{6} \int_{D_{min}}^{D_{max}} D^3 N(D) \mathrm{d}D. \tag{28}$$

The $D_0$ is the raindrop diameter when the following equation is satisfied (Fukao and Hamazu, 2014):


$$\frac{0.001\pi\rho_w}{6} \int_{D_{min}}^{D_0} D^3 N(D)\mathrm{d}D = \frac{0.001\pi\rho_w}{6} \int_{D_0}^{D_{max}} D^3 N(D)\mathrm{d}D. \tag{29}$$

According to Eq. 29, $D_0$ is the raindrop diameter that divides the total volume of raindrops into two equal halves. For the integral calculations in Eqs. 28–29, $D_{min}$, $D_{max}$, and the interval segments of $D$ were set as in the scattering simulation in Section 3.4 (0.01 mm, 9 mm, and 1024 segments, respectively). For Eq. 29, $D_0$ was obtained by iteration when the difference between the left and right sides was minimised. $N_w$ was obtained by substituting $L_{WC}$ and $D_0$ estimated from Eqs. 28–29 into

Eq. 27. In this study, Eqs. 21–27 were applied to the TANX data and Eqs. 27–29 to the NUX data.





### 3.6 Estimation and comparison of dominant cloud microphysical processes

In this study, we attempt to infer the dominant cloud microphysical processes from the vertical variations of multiple
cloud physical quantities derived from vertical pointing observations. For example, if there is no vertical variation in $L_{WC}$,
i.e., no condensation or evaporation, and $D_0$ increases (decreases) downward, collision-coalescence (breakup) is expected to
be dominant. As Misumi et al. (2021) pointed out, $N_w$ is likely to increase as the number of small raindrops increases. Based
on this idea, $N_w$ should decrease (increase) downward if collision-coalescence (breakup) is predominant because the number
concentration of small raindrops should decrease (increase). If condensation (evaporation) is dominant, $L_{WC}$ increases
(decreases) downward (e.g., Williams, 2016). When condensation (evaporation) is dominant, the number of small raindrops
increases (decreases), so the value of $N_w$ is expected to become larger (smaller). Notably, Williams (2016) similarly
estimated the dominant cloud physical processes from vertical changes in cloud physical quantities; however, the difference
between this study and Williams (2016) is that the latter used the total number of drops ($N_t$) instead of $N_w$. As $N_w$ is used in
our discussion, it is easier to discuss the behaviour of smaller raindrops. This study enables more robust discussion by
confirming vertical variations in multiple cloud physical quantities.

The dominant cloud microphysical processes estimated from the vertical pointing observations were compared using
TANX polarimetric parameters following the method of Kumjian and Prat (2014). Kumjian and Prat (2014) proposed a
method to estimate the predominant warm cloud microphysical processes from the altitude variation of polarisation
parameters. For example, if both the $Z$ and $Z_{DR}$ values increase (decrease) toward the lower layers, coalescence (breakup)
should be dominant. We used only the PPI data of TANX at elevation angles of 2.5°, 3.7°, and 4.9° that are observed above
the Shigaraki MU Observatory.

## 4 Results and discussion

### 4.1 Overview

This study focuses on a rainfall event that occurred on 2 June 2023 at the Shigaraki MU Observatory. Figure 7a–c and
Figure 7d–e are time–height cross sections of parameters obtained from NUX and MU radar observations, respectively. The
tropospheric observation mode of the MU radar was from 09:40 to 16:59 JST (UTC+9) on this day. As the 0 °C altitude on
this day was roughly 4900 m according to the ERA5 data, only data at an altitude of about 3400 m or lower (liquid phase)
were used. The black dots in Figure 7 indicate the times that satisfy the criteria for typical stratiform rainfall defined in
Section 3.2 and were used in the following analyses. Figures 7d and 7e show that stratiform precipitation was dominant,
particularly during the second half of the MU radar observation period. Figure 7c shows that the reflectivity in typical
stratiform rainfall was generally around 30 dBZ and occasionally exceeded 40 dBZ. In Figure 7a, Doppler velocity values



are negative for most of the time, and temporal fluctuations in the values can also be observed in the liquid phase. Figure 7b shows that during periods of typical stratiform rainfall, the fluctuations in $\sigma_d^{NUX}$ are smaller compared to other periods. Note that the data shown in Figure 7a–c correspond to the short pulse region.




Figure 7. Time–height cross sections of (a) $\overline{V_d^{NUX}}$, (b) $\sigma_d^{NUX}$, (c) $Z^{NUX}$, (d) $\overline{V_a^{MU}}$, and (e) $\sigma_a^{MU}$ for the analysis period, 09:40 to 16:59 JST (Japan Standard Time, UTC+9), on 2 June 2023. The black dots at the bottom of panels (a–e) indicate times when the criteria for "typical stratiform rainfall" defined in Section 3.2 are satisfied. The shading in (a–c) shows only the periods for which vertical pointing observations were performed. The shading in (a-b) and (d-e) indicates only the specific altitude data used for the analysis. In (a) and (d), positive values represent upward motion and negative values represent downward motion. The grey horizontal line in (c) indicates the maximum beam altitude over the Shigaraki MU Observatory when observed by TANX at an elevation angle of 4.9°.

We use TANX data for comparison with vertical observations. The TANX data used in the following analysis include only the data from times when typical stratiform rainfall was identified by the MU radar for at least 4 min out of 5 min. In this analysis, 52 TANX volume scan datapoints were used. Figure 8 shows time-height cross sections of TANX data just above the Shigaraki MU Observatory. Note that only the data from the lower three layers in Figure 8 are used in this study.





The temporal and altitude variations in radar reflectivity are generally consistent between Figures 7c and 8a; for example, both show relatively strong radar reflectivity around 14:40. In some cases, the temporal and vertical variations of $Z_{DR}^{TANX}$ and

$K_{DP}^{TANX}$ correspond to changes in $Z^{TANX}$; for example, around 14:40, both $Z_{DR}^{TANX}$ and $K_{DP}^{TANX}$ tend to increase as $Z^{TANX}$ increases (Figure 8). Notably, from 13:00 to 14:00, the high-reflectivity area interpreted as the melting layer decreased in altitude to around 3500 m (Figure 8a). However, vertical observations show no indication that the melting layer altitude decreased to around 3500 m (Figure 7c). Instead, considering the beam broadening of TANX, the beam at an elevation angle of 4.9° may intermittently capture the lower edge of the melting layer (Figure 7c), potentially causing an enhancement in

radar reflectivity (Figure 8a). Data at an elevation angle of 4.9° are used throughout this analysis, although they might be affected by the melting particles.

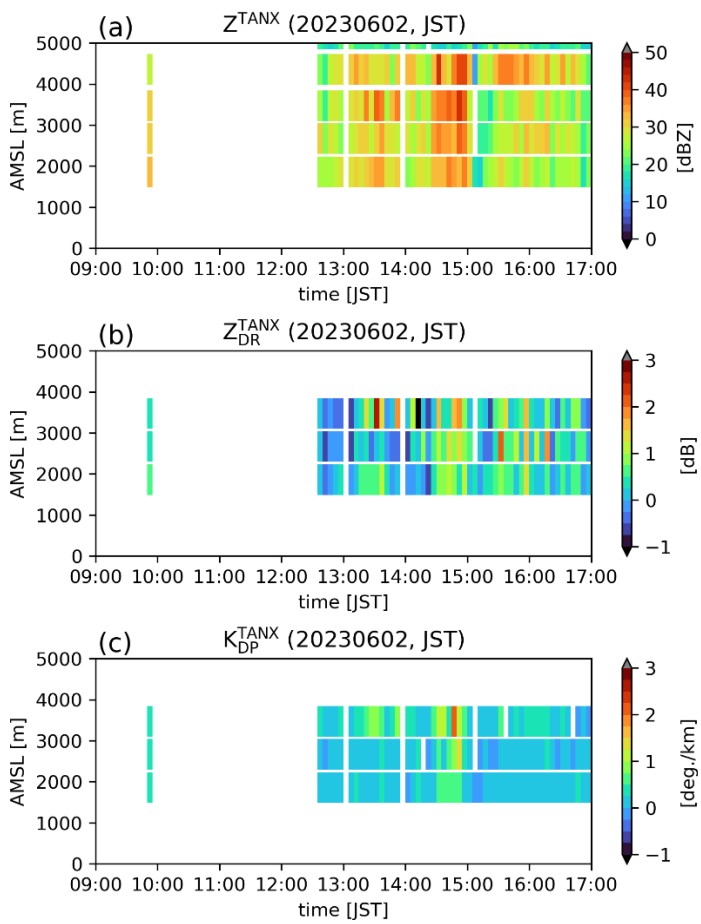

Figure 8. Time–height cross sections of (a) $Z^{TANX}$, (b) $Z_{DR}^{TANX}$, and (c) $K_{DP}^{TANX}$ for the analysis period, 09:40 to 16:59 JST (Japan Standard
Time, UTC+9), on 2 June 2023. The data are processed as PPI data. Data are from the lower layer at elevation angles of 2.5, 3.7, 4.9, and 7.5 degrees. Note that the shading in (b-c) indicates only the specific altitude data used for the analysis (only 2.5, 3.7, and 4.9 degrees).



## 4.2 Retrieval of DSD parameters

Figure 9 shows the contoured frequency by altitude diagrams (CFADs; Yuter and Hauze, 1995) of the DSD parameters at
the times identified as typical stratiform rainfall. Figure 9d exhibits a decreasing number of grid points at higher altitudes,
which is due to the effect of variations in the 0 °C altitude between 09:40 and 16:59 JST. Figures 9a and 9b show that the
values of $\mu$ and $\Lambda$ generally increase with decreasing altitude except for the highest altitude; the median of $\mu$ changes from
$-0.31$ at 3160 m to 1.21 at 1660 m, and the median of $\Lambda$ changes from 3.34 to 4.22. This trend is the same as that described
by Rao et al. (2006) and Kirankumar et al. (2008) for stratiform precipitation and is also consistent with the trend of the $\mu$–$\Lambda$
relationship derived in previous studies (e.g., Rao et al., 2006; Wen et al., 2020). The present case has smaller median values
for both $\mu$ and $\Lambda$ than those reported by Rao et al. (2006) and Kirankumar et al. (2008), suggesting that this case has a wider
range of raindrop sizes; i.e., it contains a variety of raindrops from small to large in diameter. $N_0$ is a parameter whose units
vary with the value of $\mu$, and Figure 9c shows that $N_0$ increases as $\mu$ increases toward lower altitudes except for the highest
altitude.


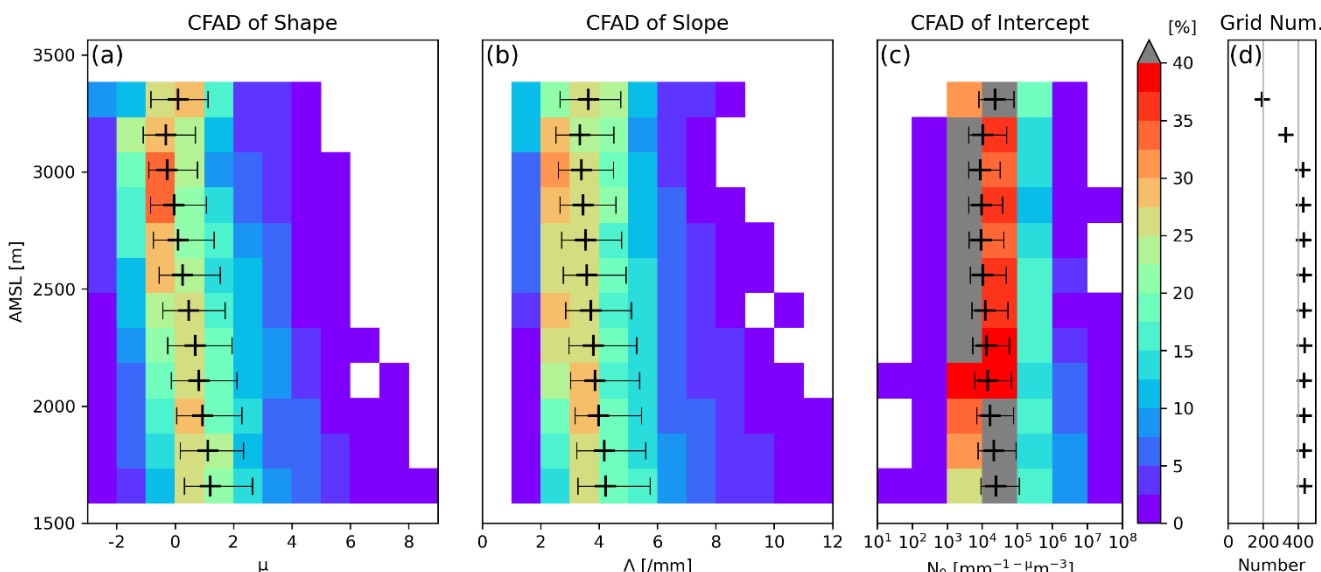

Figure 9. CFADs for (a) $\mu$, (b) $\Lambda$, and (c) $N_0$ at the times determined to be typical stratiform rainfall. The + marks and bars in (a–c) indicate the median and interquartile range, respectively, for each altitude. The + marks in (d) indicate the number of grid points analysed for each altitude. The vertical interval is 150 metres for (a) to (d).


Figure 10 shows the DSD using the median values of $\mu$, $\Lambda$, and $N_0$ for each altitude shown in Figure 9. The number
concentrations of both small and large particles tend to decrease with decreasing altitude (Figures 10a and 10d). On the other
hand, the number concentration of intermediate size particles (around 1.5 mm) increases with decreasing altitude (Figures
10b and 10c). Thus, the width of the DSD tends to narrow as the altitude decreases in this case.




Figure 10. (a) DSD using the median values of $\mu$, $\Lambda$, and $N_0$ for each altitude; (b–d) enlarged views of the diameter ranges (b) 1.0–1.5 mm, (c) 1.5–2.0 mm, and (d) 3.0–3.5 mm. The vertical axes are (a) logarithmic and (b–d) linear. For better visualisation, the altitude levels are thinned out.


## 4.3 Comparison of the retrieved cloud physical quantities

Figure 11 presents the altitude variation of the estimated median $D_0$, $L_{WC}$, and $N_w$ retrieved from the NUX and MU radar vertical observations and from TANX PPI scans. Notably, the data from the vertical observations are every 30 s, while the TANX conducts one volume scan every 5 min. Figure 11a shows that the $D_0$ estimated from NUX and MU radar observations (magenta + marks; Eq. 29) was generally within the range of $D_0$ estimated using $Z_{DR}$ (Eqs. 21–23), although it was slightly larger than that estimated by the method of Kim et al. (2010) below 2300 m. Figure 11b shows similarity in the vertical variation trends between the $L_{WC}$ estimated from NUX and MU radar vertical observations (magenta + marks; Eq.




28) and that from TANX using $K_{DP}$ (blue dots; Eq. 26), although the former exhibits larger values. In addition, the $N_w$

estimated from vertical observations (magenta + marks; Eq. 27) and that using $L_{WC}$ estimated from $K_{DP}$ (blue dots) roughly agreed (Figure 11c). The slight discrepancy in $N_w$ above 2600 m (Figure 11c) could be attributed to the fact that $D_0$ estimated by vertical observation is smaller than that estimated when the method described by Kim et al. (2010; Eq. 22) was used to calculate $N_w$ (blue dots; Eq. 27). However, the $D_0$ estimated from the vertical observations is close to that estimated by the method of Anagnostou et al. (2008; Eq. 21) above 2600 m, suggesting that its fluctuations can be estimated with reasonable accuracy.


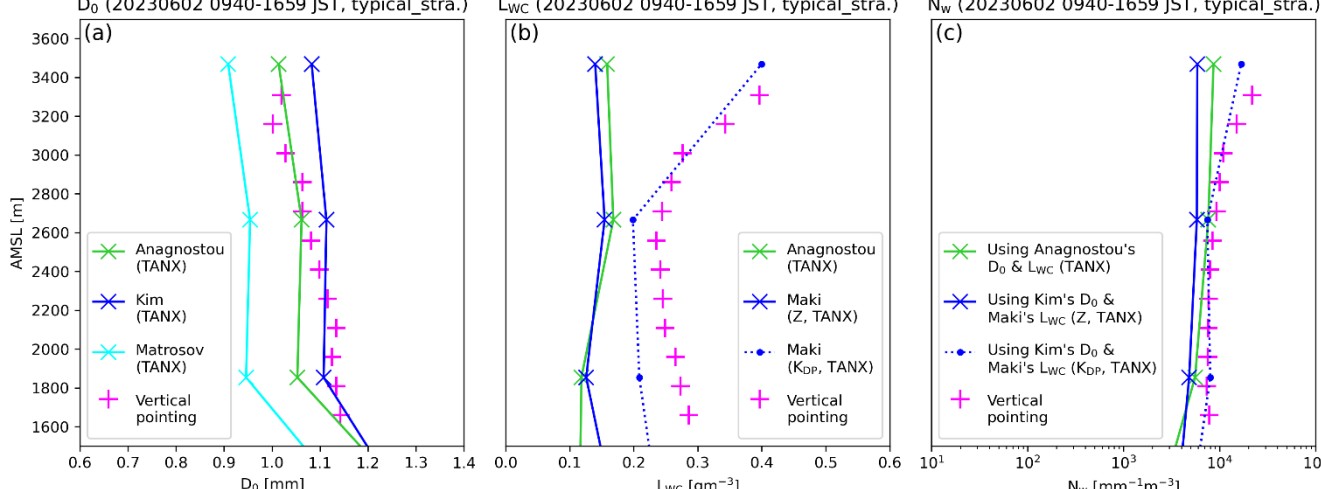

Figure 11. Comparison of the median values for each altitude of the parameters (a) $D_0$, (b) $L_{WC}$, and (c) $N_w$. The green, blue, and cyan marks are parameters derived from Eqs. 21–27 using TANX PPI data. The magenta + marks are parameters derived from Eqs. 27–29 using vertical observation data.


Maki et al. (2005) showed the error range of Eq. 26 (blue dots in Figure 11b) using the normalised mean absolute error, indicating that when $L_{WC}$ is around 0.25 g/m³, the error is approximately ±0.07 g/m³. The absolute difference between the median $L_{WC}$ estimated from the $K_{DP}$ of TANX and that estimated from the vertical observations (linearly interpolated to the TANX altitudes) is about 0.06 g/m³ at 1850 m AMSL and about 0.05 g/m³ at 2660 m AMSL. This means that the difference

between the $L_{WC}$ estimated from vertical observations and that estimated using the $K_{DP}$ of TANX is within the error range specified in Eq. 26. Furthermore, when $K_{DP}$ is simulated using the T-matrix method with DSD parameters obtained from vertical observations, the median values of $K_{DP}$ are 0.11 °/km at both 1810 m and 2710 m (Figure 12). The median $K_{DP}$ value for the nearest grid point using TANX is also 0.11 °/km at both heights. Thus, the difference between the DSD used by Maki et al. (2005) and the DSD retrieved in this study may be considered to cause the discrepancy in the estimated $L_{WC}$.




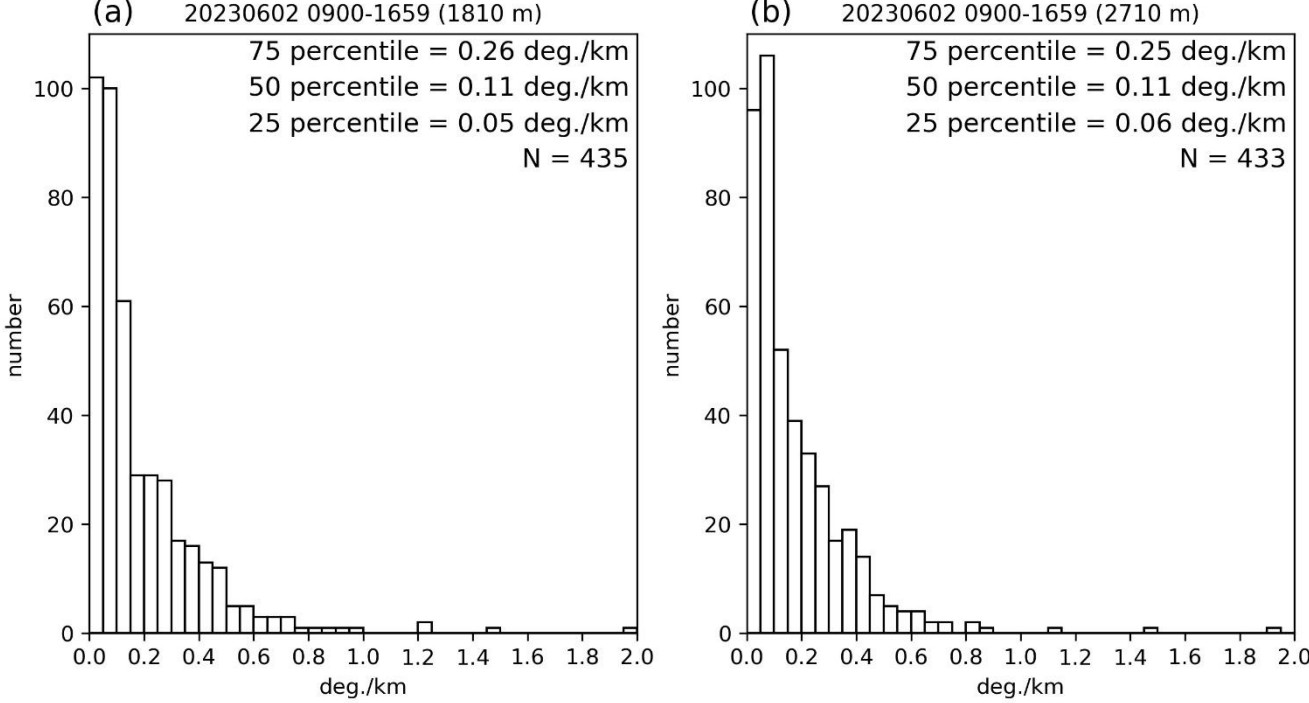

Figure 12. Histograms of $K_{DP}$ calculated using the T-matrix method from DSD estimated from vertical observations, (a) at 1810 m and (b) at 2710 m. N is the total number of samples.

$L_{WC}$ estimated from $Z^*$ (Eqs. 24–25) differed significantly from that estimated using vertical observations (Eq. 28) and $K_{DP}$ (Eq. 26; Figure 11b), not only in its values but also in its vertical variation trend. The sensitivity of $Z^*$ and $K_{DP}$ to DSD variations is discussed. The $n$th order moment of DSD is defined by the following equation (e.g., Fukao and Hamazu, 2014),

$$m_n \equiv \int_0^\infty D^n N(D) \mathrm{d}D \,. \tag{30}$$

Assuming the Rayleigh scattering approximation, $Z^*$ is the 6th moment of DSD from Eq. 2 and $K_{DP}$ is the 4.3–4.9 moment

(Ryzhkov and Zrnic, 2019). On the other hand, $L_{WC}$ is represented by the 3rd moment from Eq. 28. Thus, $Z^*$ is a higher-order moment than $K_{DP}$ and has higher sensitivity to DSD variations, making $L_{WC}$ estimations derived from $Z^*$ susceptible to errors due to DSD variations.

    In addition, $Z$ and $Z_{DR}$ are considered to be affected by attenuation, although the attenuation correction of $Z^{TANX}$ and $Z_{DR}^{TANX}$ is performed using $K_{DP}^{TANX}$ following Maesaka et al. (2011). Notably, horizontal specific attenuation ($A_h$ [dB/km]) is

less sensitive to raindrop shape than specific differential attenuation ($A_{DP}$; Ryzhkov and Zrnic, 2019). This means that $A_h$ is also more susceptible to the effects of small raindrops that are not sensitive to $K_{DP}$. Consequently, the attenuation correction for $Z^{TANX}$ is especially likely to be underestimated in weak rainfall, and the error may increase over about 38 km of rain path.




As $A_h$ is more susceptible to variations in DSD than $A_{DP}$ (Ryzhkov and Zrnic, 2019), the error in $Z$ is likely to be larger than that in $Z_{DR}$.

Overall, the $D_0$, $L_{WC}$, and $N_w$ estimated from the vertical observations in this study are broadly consistent with those estimated using $K_{DP}$ and $Z_{DR}$ (Figure 11). Therefore, the cloud physical quantities estimated from vertical observations can be considered reliable.

### 4.4 Confirming the influence of $\sigma_d^{NUX}$ broadening

We discuss the broadening effect on $\sigma_d^{NUX}$. Figure 13 shows the differences in DSD parameters and cloud physical quantities estimated at each altitude with and without the removal of the influence of $\sigma_{broadening}$. We focus particularly on μ. The mean absolute error (MAE) for μ is 0.28 (Table 4). As shown in Figure 13a, the values of the interquartile range (IQR) of μ vary between 2 and 3 at all altitudes. The MAE represents about 10% of the IQR, and since the Wilcoxon test revealed significant differences in all layers, systematic shifts in the distribution are indicated. Given that μ is a sensitive shape

parameter governing the curvature of the DSD, a systematic bias equivalent to 10% of the natural variability can lead to non-negligible errors in derived microphysical properties. The statistical significance further confirms that this correction removes a persistent bias, thereby improving the quantitative accuracy of the retrieval. Therefore, even during the typical stratiform rainfall period defined in this study, $\sigma_d$ correction remains important.

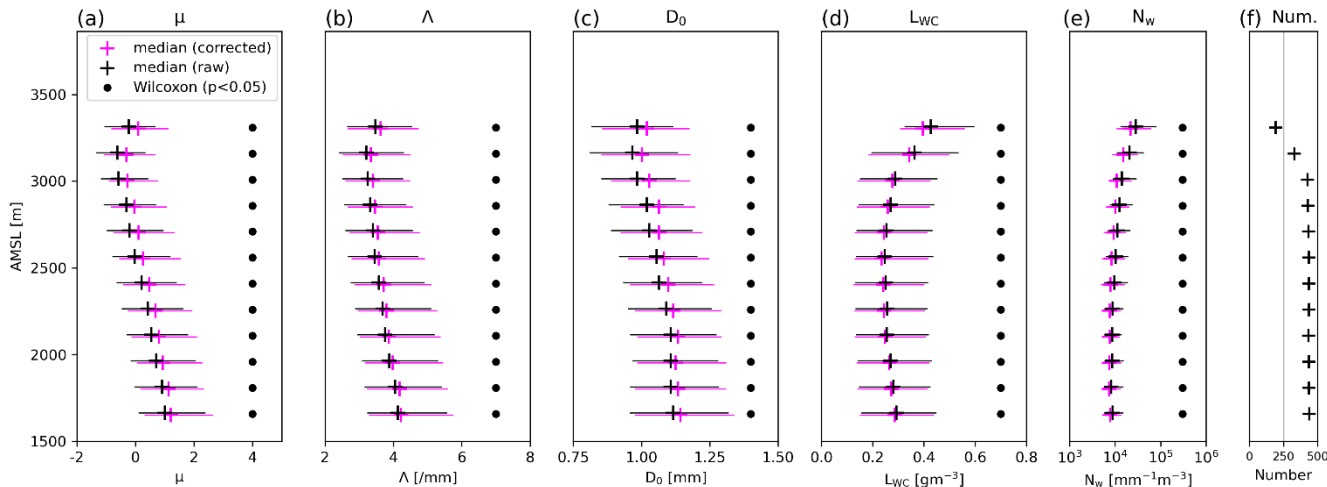


Figure 13. Comparison of (a–b) DSD parameters and (c–e) cloud physical quantities estimated (magenta) with the correction for the influence of $\sigma_{broadening}$ and (black) without the correction. Note that the correction using $\overline{V_a^{MU}}$ is performed for both magenta and black. (f) indicates the sample size. For each altitude, the + marks indicate the median, and the bars represent the interquartile range (IQR). Black circles at each altitude indicate significant differences (p < 0.05) detected by the Wilcoxon signed-rank test.






Table 4. Mean absolute error (MAE) for each parameter when comparing cases with and without correction for the influence of $\sigma_{\text{broadening}}$ and $\overline{V_a}$.

| Parameter | MAE ($\sigma_{\text{broadening}}$) | MAE ($\overline{V_a}$) |
|---|---|---|
| $\mu$ [dimensionless] | 0.28 | 0.24 |
| $\Lambda$ [mm$^{-1}$] | 0.15 | 0.24 |
| $D_0$ [mm] | 0.03 | 0.06 |
| $L_{\text{WC}}$ [g/m$^3$] | 0.01 | 0.06 |
| $N_w$ [mm$^{-1}$m$^{-3}$] | $6.4 \times 10^3$ | $2.9 \times 10^4$ |

**4.5 Confirming the influence of $\overline{V_a}$**

This section evaluates the influence of $\overline{V_a}$ on DSD estimation. Figure 14 shows the differences in DSD parameters and cloud physical quantities estimated at each altitude when the influence of $\overline{V_a}$ is removed, as in Eq. 18, or when assuming $\overline{V_{\text{tz}}^{\text{NUX}}} = \overline{V_d^{\text{NUX}}}$. Figure 14 indicates that significant differences are present in Wilcoxon's signed-rank test for all parameters and at almost all altitudes. The MAE for both $\mu$ and $\Lambda$ is 0.24 (Table 4), which corresponds to approximately 10% of the IQR

(Figure 14). Furthermore, the influence of $\overline{V_a}$ correction on $D_0$, $L_{\text{WC}}$, and $N_w$ evaluated by MAE is larger than that of $\sigma_d$ correction. This fact demonstrates that even during typical stratiform rainfall characterised by a small $|\overline{V_a}|$ ($|\overline{V_a^{\text{MU}}}| \leq 1.0$ m/s), $\overline{V_a}$ has a significant influence on the estimation of DSD parameters and cloud physical quantities. Therefore, when estimating DSD from vertical observations, the retrieval of $\overline{V_a}$ needs to be performed carefully and accurately.






Figure 14. The same as Figure 13, but comparing estimates with $\overline{V_a}$ correction using $\overline{V_a^{MU}}$ and those without the correction indicated by dark red or blue marks. The panels show the cases for (a–f) when $\overline{V_a^{MU}} \geq 0$, i.e., upward air motion, and for (g–l) when $\overline{V_a^{MU}} < 0$, i.e., downward air motion. Note that the $\sigma_d$ correction is performed in all estimates. (f) and (l) indicate the sample size, and only cases with 100 or more samples per altitude are plotted in all panels.

Based on the discussions in this section and Section 4.4, even in typical stratiform rainfall, the estimation of DSD is susceptible to the effects of vertical air motion, horizontal wind, and turbulence. Therefore, accurate DSD estimation without raw Doppler spectra requires data correction as performed in previous studies (e.g., Shupe et al., 2008; Williams et al., 2016; Pang et al., 2021) and this study. This fact is considered applicable to the Ku-band satellite-borne Doppler radar, which operates at a frequency close to that of X-band radar, suggesting that accurate DSD estimation using pulse-pair Doppler processing could be achieved if vertical air motion is accurately retrieved and horizontal wind is derived using along-track and cross-track techniques.





### 4.6 Estimation and validation of cloud microphysical processes derived from vertical pointing observations

The estimated DSD parameters and cloud physical quantities vary with altitude (Figures 9 and 11), and the cloud microphysical processes responsible for their variations are discussed. At altitudes between 3010 m and 1660 m, the overall variation in $L_{WC}$ values derived from vertical pointing observations is relatively small (0.24–0.29 g/m$^3$; Figure 11b). This indicates that condensation and evaporation are minimal at these altitudes. Between altitudes of 3010 m and 2110 m, $D_0$ tends to increase (from 1.00 to 1.13 mm) and $N_w$ tends to decrease (from 10900 to 7600 mm$^{-1}$m$^{-3}$) toward the lower altitudes (Figures 11a and 11c), suggesting that the collision-coalescence of raindrops is predominant. At altitudes below 2110 m, the changes in $D_0$ and $N_w$ with altitude are slight, with variation in $D_0$ from 1.13 to 1.14 mm and the range of $N_w$ being from 7300 to 7800 mm$^{-1}$m$^{-3}$. This may be approaching the equilibrium state of DSD, which is characterised by simultaneous coalescence and breakup due to collisions among raindrops (Zawadzki and De Agostinho Antonio, 1988). Notably, equilibrium DSD is often expressed as a bimodal (e.g., McFarquhar, 2004; Okazaki et al., 2023; Unuma, 2024) or trimodal (e.g., Chen and Lamb, 1994) distribution, and such distributions cannot be expressed using a gamma distribution. However, vertical one-dimensional numerical simulations by Barthes and Mallet (2013) have shown that $\mu$ and $\Lambda$ tend to increase downward as equilibrium state is approached through collision-coalescence and breakup. The DSD in this study (Figures 9 and 10) may also be approaching equilibrium state downward. This is because, despite median $\mu$ and $\Lambda$ increasing below an altitude of 2110m (from 0.81 to 1.21 and from 3.86 to 4.22 mm$^{-1}$, respectively), the changes in $D_0$ and $N_w$ are minimal (Figures 11a and 11c). At altitudes above 3010 m, $L_{WC}$ and $N_w$ obtained from vertical pointing observations decrease downward (Figures 11b and 11c), suggesting that evaporation of small raindrops may be dominant. At these altitudes, $D_0$ increases or decreases slightly (Figure 11a). In particular, a decreasing $D_0$ is not the behaviour generally expected in conditions where evaporation is predominant (Xie et al., 2016). However, Xie et al. (2016) also show that special cases exist in which $D_0$ decreases downward due to evaporation in the region where the initial $\mu \approx 0$. As the median value of $\mu$ at the highest altitude analysed in this case is 0.10 (Figure 9a), these special fluctuations in $D_0$ might have been caused by evaporation. Therefore, in this analysis, collision-coalescence is considered to have contributed the most, with breakup and evaporation also contributing.

The verification of the dominant cloud microphysical processes inferred from vertical observations is performed using TANX data. Figure 15 shows the predominant cloud microphysical processes in the liquid phase estimated from the method of Kumjian and Prat (2014) using TANX data. From Figure 15, collision-coalescence and breakup are dominant most of the time, followed by signals of evaporation and size-sorting. This is generally consistent with the dominant cloud microphysical processes estimated from vertical observations. This finding shows that dominant cloud microphysical processes can be estimated from the altitude variations of multiple cloud physical quantities derived from DSD parameters retrieved from vertical observations.



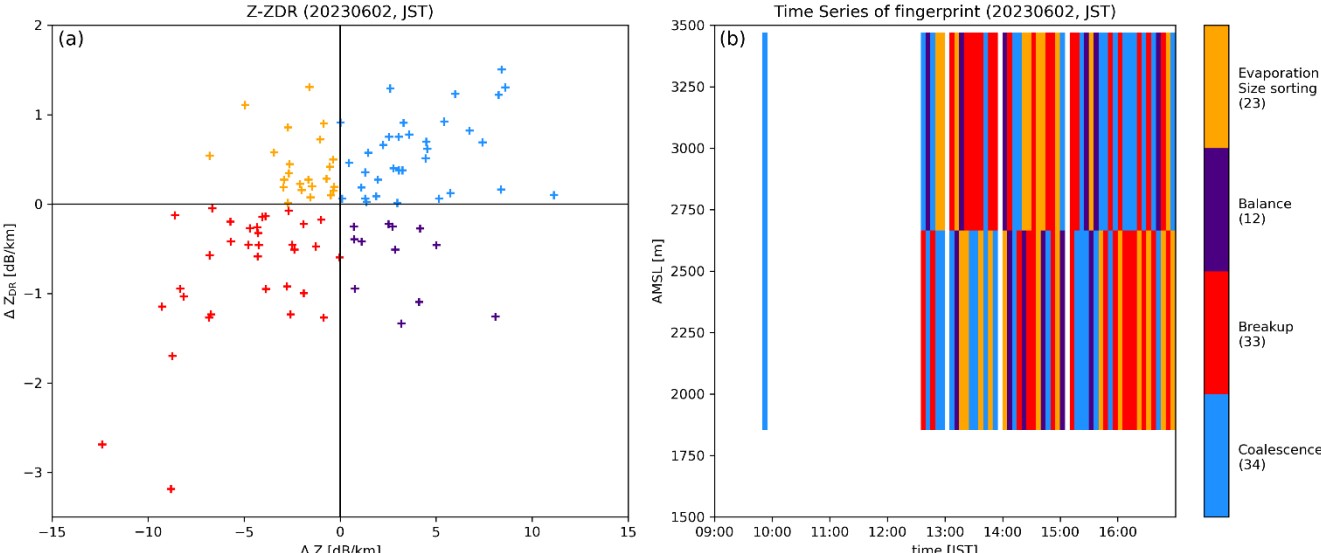

Figure 15. The fingerprints of cloud microphysical processes in warm rain are shown in (a) the $\Delta Z$–$\Delta Z_{DR}$ plane with decreasing altitude, and (b) the time-height cross section. The colours of the plots and shades correspond to the legend, and the numbers in the legend represent the number of data points for each category.

From the above discussion, the altitude variations of $\mu$ and $\Lambda$ in Figure 9 are primarily attributed to collision-coalescence and breakup. This perspective differs from the factors described by Rao et al. (2006) and Kirankumar et al. (2008) for DSD altitude variations in the tropics. This difference should be related to the relative humidity below the melting layer. Kirankumar et al. (2008) inferred a significant contribution from evaporation based on the results of numerical simulations (Ferrier et al., 1996), which indicate that evaporation contributes to continental precipitation at altitudes below the melting layer. However, in this analysis, the relative humidity in the ERA5 reanalysis data exceeded 75% for most of the period at the analysed altitude (Figure 16), indicating that the area below the melting layer is moist (e.g., Nishii et al., 2025). These moist conditions should have prevented the evaporation of the small-sized raindrops and increased the opportunities for collision-coalescence and collision-breakup between small and large raindrops.



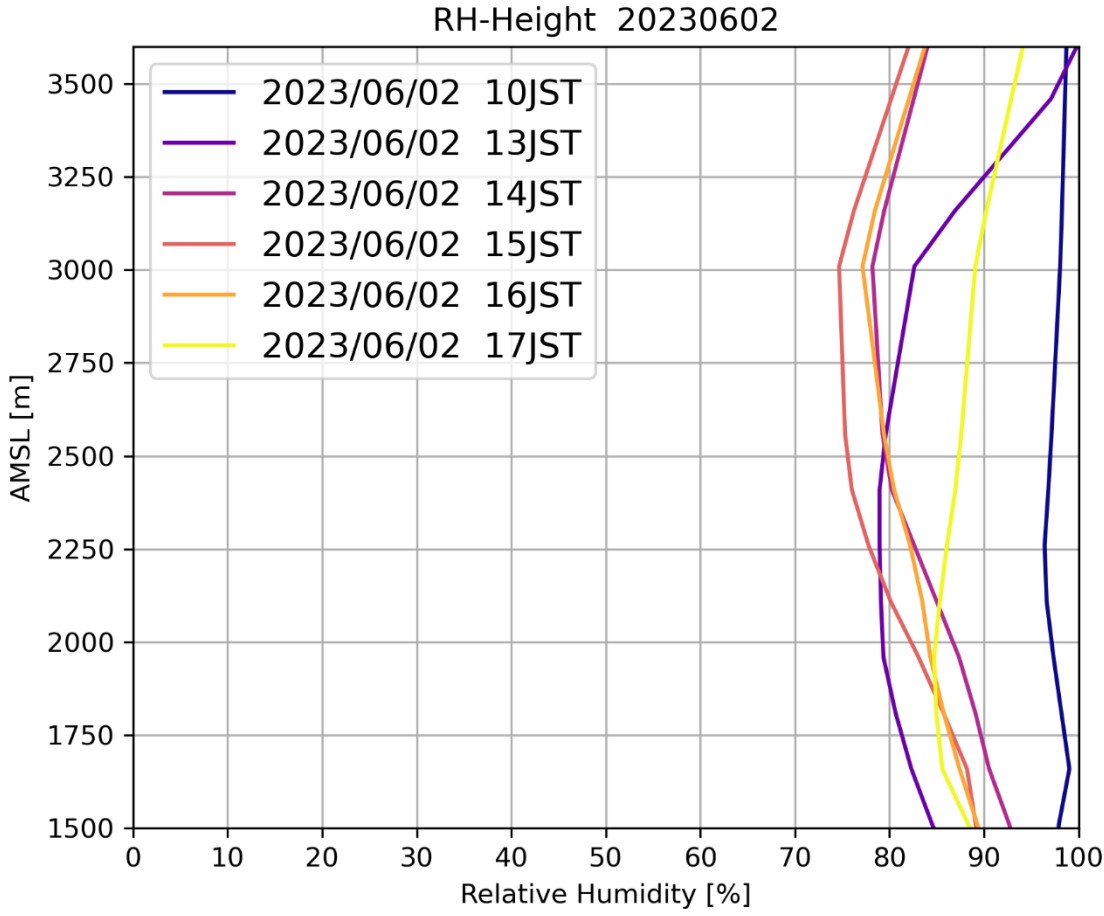

Figure 16. Vertical profiles of relative humidity over the Shigaraki MU Observatory are shown at each time. Only the data closest to the time depicted in the fingerprint in Figure 15b are presented.

## 5. Summary

625

In this study, we introduce a method to estimate the vertical profile of DSD parameters and retrieve cloud physical quantities using vertical pointing observation data from X- and VHF-band radars, and apply it to a typical stratiform rainfall case in Japan. As NUX does not output raw spectra, the DSD estimation method proposed by Fukao and Hamazu (2014) utilising Doppler spectral width $\sigma_d$ was used. Additionally, it was modified to account for non-Rayleigh effects using T-

630

matrix method simulations. Even during periods of typical stratiform rainfall, contamination by $\overline{V}_a$ and $\sigma_{broadening}$ in the estimation of DSD parameters and cloud physical quantities is shown to be non-negligible. This indicates that even under the calmest stratiform rainfall conditions, accurate DSD estimation requires correction for the effects of vertical air motion,



horizontal wind, and turbulence. Additionally, this fact might indicate the challenges involved in applying DSD analysis to satellite Doppler radar. However, for rainfall that satisfies the conditions for typical stratiform rainfall defined in this study, the broadening effect of $\sigma_d$ due to wind shear can be neglected even with radar having a vertical resolution of approximately 150 metres. We also confirm that the cloud physical quantities retrieved using the DSD estimated from vertical observations are reasonably accurate.

The $\mu$ and $\Lambda$ estimated from vertical pointing observations generally increase with decreasing altitude. The estimated vertical variations in cloud physical quantities ($D_0$, $L_{WC}$, and $N_w$) suggest that collision-coalescence is the main factor causing vertical variations in these DSD parameters, with some influence from breakup and evaporation. The dominant cloud microphysical processes were also confirmed by applying the method of Kumjian and Prat (2014) to TANX data, and were generally consistent with vertical pointing observations. These results indicate that the contributing cloud microphysical processes are different, although the altitude variations of $\mu$ and $\Lambda$ are similar to those in previous studies in the tropics (Rao et al., 2006; Kirankumar et al. 2008). The moist environment probably prevented the evaporation of small raindrops in this study.

This study demonstrates that combining the vertical variations of multiple cloud physical quantities enables the estimation of dominant cloud microphysical processes. We conclude that accurate estimation based solely on the vertical variation of DSD parameters or that of one cloud physical quantity, as performed in previous studies, is difficult. As changes in cloud physical quantities directly reflect cloud microphysical processes, we show that the accurate estimation of cloud microphysical processes can be achieved by combining multiple vertical variations in cloud physical quantities, such as $D_0$, $L_{WC}$, and $N_w$. This study suggests that significant advances in the understanding of cloud microphysics can be achieved by interpreting the vertical variations in estimated DSD parameters in the context of the dominant cloud microphysical processes using only data from vertical pointing observations.

The vertical observations of NUX at the Shigaraki MU Observatory were performed until November 2025. This method will be used to conduct statistical analyses of the DSD for the rainy season in Japan.

**Code and data availability**

The analysis in this study was conducted using Python3, C, and fortran90 scripts developed by the authors. These scripts were specifically used for the different analyses and assessments presented in the paper. Due to their length, the number of scripts, and the current lack of documentation suitable for public release, the code has not been deposited in a public repository. However, the authors can provide the scripts upon reasonable request for academic and research purposes. Interested researchers can contact the corresponding author (YG) at [ygoto@nagoya-u.jp](mailto:ygoto@nagoya-u.jp). The following Python modules are available online: the PyTmatrix code (https://github.com/jleinonen/pytmatrix) and the PyXRAIN code for analysing XRAIN data (https://github.com/YuukiWada/PyXRAIN).

The TANX dataset (https://search.diasjp.net/ja/dataset/MLIT_XRAIN) and ERA5 hourly data on pressure levels (https://cds.climate.copernicus.eu/datasets/reanalysis-era5-pressure-levels?tab=download) are also available online. The NUX and MU radar data are available from the corresponding author (YG) upon request.

**Author contribution**

Data analyses were performed by YG and NT. Writing and revising were conducted by YG and TS. Conceptualisation of the method and interpretation were shared between YG, TS, NT, HH, and SS. The MU Radar data were provided by HH. NUX was maintained by HM and MK.

**Competing interests**

The authors declare that they have no conflict of interest.

**Acknowledgements**

This work was performed using NUX of the Institute for Space-Earth Environmental Research, Nagoya University. The MU radar belongs to and is operated by the Research Institute for Sustainable Humanosphere (RISH), Kyoto University.
TANX is operated by The Ministry of Land, Infrastructure, Transport, and Tourism, Japan, and that data are collected and provided by the Data Integration and Analysis System (DIAS), developed and operated by the Ministry of Education, Culture, Sports, Science, and Technology, Japan.

**Financial support**

This work was supported by JSPS KAKENHI [Grant Number JP22H00177]. This work was also financially supported by JST SPRING [Grant Number JPMJSP2125]. The first author (YG) would like to take this opportunity to thank the "THERS Make New Standards Program for the Next Generation Researchers."



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
