# Peer review of "Estimation of vertical profiles of raindrop size distribution and cloud microphysical processes in stratiform rainfall using vertical-pointing X- and VHF-band radars"

_EGUsphere, 2025_

## Author Comment (AC1)

Title: Estimation of vertical profiles of raindrop size distribution and cloud microphysical processes in stratiform rainfall using vertical-pointing X- and VHF-band radars

Authors: Yusuke Goto, Taro Shinoda, Haruya Minda, Moeto Kyushima, Hiroyuki Hashiguchi, Nozomu Toda, and Shoichi Shige

https://doi.org/10.5194/egusphere-2025-5944

Journal: Atmospheric Measurement Techniques

We greatly appreciate all your comments, instructions, and suggestions for our paper. Our response is written in blue.

**General comments:**

The authors proposed a method for estimating the vertical distributions of DSD parameters and cloud microphysics-related parameters for layered rainfall events using vertically pointing observations from X-band and VHF-band radars. Although the validation was limited to specific regions and cases, their analysis was conducted with great care and is considered highly valuable. However, whilst the technical steps undertaken are listed in the main text, the reasons for undertaking them and their advantages are generally absent, and the relationships between them remain unclear, rendering the text very difficult to read. Consequently, the novelty of this research is hard to discern. There must be pros and cons associated with each radar used; these should be explicitly stated whilst reviewing relevant prior research.

Furthermore, the advantages gained from combining these radars and how they differ from previous research should be clearly stated in the introduction. Relatedly, within the methodology section, it would be preferable to include a brief explanatory sentence at the beginning of each subsection to clarify the context within the text.

Thank you for your comments. We acknowledge that our explanation regarding the rationale and advantages of the correction methods was insufficient. We also agree that the logical flow of the text and the advantages of the radar system were not fully addressed. In the revised manuscript, we will carefully review the text to clarify these points and improve the overall readability.

**Specific comments:**

1. L34-63: This section should outline what is known from ground-based drop-size distribution observations and their limitations, and subsequent sections should highlight the importance of the vertical distribution of cloud microphysical properties.

    Based on your feedback, we will revise the content to emphasize the limitations of ground-based observations and the importance of estimating DSD aloft. In particular, we will modify the text with a focus on improving the connection between paragraphs.

2. L66-67: Please briefly state why the Rayleigh approximation allows the average vertical flow to be negligible.

This is a portion of the main text, which states: "if the Rayleigh scattering approximation is valid and mean vertical air motion ($\overline{V_a}$ [m/s]) can be ignored." This lists two separate necessary conditions. It does not imply that valid Rayleigh scattering automatically justifies ignoring $\overline{V_a}$.

3. L-103: Having read this far, I could not discern why synchronized observations of X- and VHF-band radars are practical. As noted in the general comments, the introduction requires restructuring to state the pros/cons of X-band and VHF-band explicitly, and to clarify the advantages of combining them. Furthermore, mention should be made of why a stratiform rain event is the focus.

Thank you for your comment. The structure of the introduction in this paper is as follows: (1) Review of prior ground-based observation studies; (2) Review of prior studies on estimating DSD aloft using radar; (3) Review of prior studies on DSD estimation and cloud microphysical processes; and (4) Objectives and key points of this study. It is true that the advantages of combining X- and VHF-band radars are not mentioned until point (4); however, since they are explicitly addressed in the final paragraph of the Introduction, we believe the current structure is appropriate. Additionally, the reason for focusing on stratiform rainfall has already been stated in the final paragraph of the introduction.

Please note that based on your comment, we plan to add a sentence to the final paragraph of the introduction regarding the advantages of using X-band radar compared to S-band and C-band radars, specifically from the perspective of the backscattering cross section.

4. Section 4.1: Could you show the horizontal distribution of horizontally polarised reflectivity at the peak time for the case study? With only the vertical distribution, readers, including me, cannot fully grasp what kind of case study is being targeted.

Following your comment, Fig. Res. 1 will also be added to the main text. Fig. Res. 1 shows the 1 km-mesh synthetic radar grid point value at the 2 km level provided by the Japan Meteorological Agency at 14:40 JST. This corresponds to the time when stratiform precipitation was identified and $Z^{NUX}$ exceeded 40 dBZ. The horizontal distribution indicates values exceeding 40 dBZ above the Shigaraki MU Observatory, showing that the high-reflectivity area (> 40 dBZ) extends horizontally over the site.

[Figure]

Fig. Res. 1. Radar Reflectivity at the 2 km level at 14:40 JST, using the 1 km mesh synthetic radar grid provided by the Japan Meteorological Agency. The grey × mark indicates the location of the Shigaraki MU Observatory.

5. L587-589: It is the distribution shape that approaches equilibrium, not the state. Therefore, it would be preferable to describe 'equilibrium shape'.

   Following your comment, we will revise the description.

6. Figure 15: Why was the delta cut-off set between 2500 and 2750 m? Looking at Figs 11, 13 and 14, there appears to be no apparent kink in the vertical distributions. If there is some intention behind this, it should be explicitly stated.

   We appreciate your comment. The "cut-off" between 2500 and 2750 m in Figure 15 is not due to a physical threshold or a "kink" in the vertical profiles, but is an artifact of the observation geometry of the operational X-band radar (TANX). Unlike Figures 11, 13, and 14, which are derived from 150 m resolution vertical pointing observations (NUX and MU radar), Figure 15 is derived from TANX volume scans (PPI) at discrete elevation angles (2.5°, 3.7°, and 4.9°). The data in Figure 15 are visualized by filling the layers corresponding to these discrete elevation angles. Consequently, the apparent boundary (cut-off) around 2500–2750 m corresponds to the radar beam center of the 3.7° scan (beam width 1.2°; Table 3), rather than a physical discontinuity in the atmosphere. We will write "Using TANX data" in the revised caption of Figure 15.

7. Figure 16: Why was ERA5 data used instead of Japanese reanalysis data? Furthermore, has the accuracy been verified regardless of the reanalysis data used? For instance, near Shigaraki, I found an upper-air sounding site called Shionomisaki. I strongly recommend checking the scatter plot of relative humidity for the neighbouring grid of the reanalysis data at this site, which would enhance the reliability of this figure and strengthen the authors' arguments.

   Thank you for your comment. First, in this case, radiosonde observations were not conducted at the Shionomisaki site. According to the Japan Meteorological Agency (JMA), the automatic release of radiosondes at Shionomisaki was suspended from January 13, 2023, to May 22, 2024 (https://www.data.jma.go.jp/suishin/oshirase/pdf/20240522.pdf, in Japanese).

   Furthermore, the JMA reanalysis data are available only at 6-hour intervals. Although the JMA mesoscale objective analysis data feature a horizontal resolution of 5 km, their vertical resolution is relatively coarse (relevant levels for this study are limited to 850, 800, and 700 hPa). In contrast, while the ERA5 reanalysis data have a lower horizontal resolution, they offer a higher vertical resolution than the JMA analysis (relevant ERA5 data include levels at 850, 825, 800, 750, and 700 hPa) and provide hourly data. Since this study investigates vertical variations in DSD parameters and cloud physical quantities, we decided to use the ERA5 reanalysis data, which offer both high temporal and vertical resolutions.

   However, in this case, two radiosonde launches were conducted at the Shigaraki MU Observatory, and a comparison was made between the RH data from these launches and the ERA5 relative humidity data (Fig. Res. 2). Please note that for the left diagram in Fig. Res. 2, while the ERA5 time closest to

the radiosonde launch time is 0300 UTC, the radiosonde reaches an altitude of 1500 m at 0332 UTC. Therefore, the ERA5 time closest to the target altitude is 0400 UTC. The vertical profiles of relative humidity at target altitudes for radiosondes and ERA5 are generally consistent, although only two comparisons are available. Based on this fact, we determined that using ERA5 relative humidity data in this study is acceptable. Regarding Fig. Res. 2, since the primary objective of this study is not to verify the accuracy of ERA5, we have decided not to include this figure in the main text.

[Figure]

Fig. Res. 2. Comparison of relative humidity between radiosondes launched from Shigaraki MU Observatory and ERA5 data from the grid point horizontally nearest to the Shigaraki MU Observatory on the study date. The sonde time in the title is the launch time. ERA5 data were linearly interpolated to the altitudes of the radiosonde observations to calculate the Mean Error (ME), Mean Absolute Error (MAE), and correlation coefficient (Corr). ME, MAE, and Corr use only data from the altitudes of 1500m to 3600m shown in the figure. Radiosonde observations were sampled at a frequency of 1 Hz. Note that Japan Standard Time (JST) is UTC + 9 hours.

**Technical Corrections:** (The order of the comments has been changed.)

1. L27, 371, 374, 376, 378, 385, 390, 394, 555, 578, Table 3, Figure 11, Figure 14: $L_{WC}$ [gm$^{-3}$] should be LWC [g m$^{-3}$].

2. L29, 148: mm$^6$m$^{-3}$ should be mm$^6$ m$^{-3}$ (space required between units).

3. L35: mm$^{-1-\mu}$m$^{-3}$ should be mm$^{-1-\mu}$ m$^{-3}$ (space required between units).

4. L42, 580, Table 3, Figure 11: mm$^{-1}$m$^{-3}$ should be mm$^{-1}$ m$^{-3}$ (space required between units)

5. L67, 132, 134, 149, 556, Table 2: m/s needs to be m s$^{-1}$.

6. L185: kg/m$^3$ should be kg m$^{-3}$.

7. L271: m$^2$/s$^2$ needs to be m$^2$ s$^{-2}$.

8. L370, 371: °/km should be degree km$^{-1}$.

9. L411: UTC+9 should be UTC + 9 hours. which may apply the other place in the main text.

10. L519: dB/km should be dB km$^{-1}$.

11. Figure 9: 1/mm should be mm$^{-1}$

12. Figure 12: deg./km should be degree km$^{-1}$

Thank you for your careful and thorough feedback. We will revise the manuscript following your comments. Additionally, in response to your feedback, we will revise sections beyond the specified line numbers and figures.

13. L34: N(D) should be defined; e.g., ... a gamma distribution to show DSD (N(D)), whichi is ...

Following your comment, we will define $N(D)$ [mm$^{-1}$ m$^{-3}$] in the text.

14. L88: How large is it?

Thank you for your comment. The raindrop size at which electromagnetic resonance becomes significant varies depending on the radar frequency. For X-band radars such as NUX, the Rayleigh approximation no longer holds for raindrops with a diameter of approximately 2.5 mm, as shown in Figure 2. However, this sentence describes a general concept, so the text within the body will remain unchanged.

15. L92: ... of raindrops?

We will revise the description as follows: "the predominant cloud microphysical processes of raindrops".

16. L175: Here, N(D) should explicitly state which function is assumed.

Following your comment, we will add the sentence: "Note that N(D) is assumed to be a gamma distribution function as in Eq. 1."

These are our responses to your comments and suggestions. Thank you in advance for your kind attention.

Yusuke GOTO (ISEE, Nagoya University, Japan)

---

## Author Comment (AC2)

Title: Estimation of vertical profiles of raindrop size distribution and cloud microphysical processes in stratiform rainfall using vertical-pointing X- and VHF-band radars

Authors: Yusuke Goto, Taro Shinoda, Haruya Minda, Moeto Kyushima, Hiroyuki Hashiguchi, Nozomu Toda, and Shoichi Shige

https://doi.org/10.5194/egusphere-2025-5944

Journal: Atmospheric Measurement Techniques

We greatly appreciate all your comments, instructions, and suggestions for our paper. Our response is written in blue.

This manuscript describes a method to estimate the gamma raindrop size distribution parameters using moments from a vertically pointing X-band radar and the air motion retrieved from a VHF wind profiler. Since the X-band radar did not record the Doppler velocity spectra, the DSD parameters were retrieved from measured moments of reflectivity, mean Doppler velocity, and spectrum width. The manuscript does a nice job of describing the spectrum broadening terms (i.e., due to horizontal wind, turbulence, and shear) that contribute to the measured spectrum width, provides an error analysis, and estimates the DSD uncertainties when not including those correction terms. After clarifying a couple minor items, I believe this manuscript would be ready for publication.

We greatly appreciate such encouraging comments. We have provided our responses to each comment below.

Specific Comments:

1.      Equation (4). I do not think equations (4) and (8) are correct. From Fukao and Hamazu, the spectrum width is given by equation (5.102). The second term in equations (4) and (8) appear consistent with (5.102), but the first term is not consistent with equation (5.102). Also, to me, it does not look correct to have the mean air motion V_a(overbar) included in the spectrum width estimate. The spectrum width calculation is the width of the rain portion in the spectrum (see Fukao and Hamazu, Fig. 5.10). If V_a(overbar) is replaced with V_tz^nonRay(overbar) in equation (8), then would the spectrum width be just the first term in equation (8)? For equation (4), the text would need to define a similar variable called V_tz^Ray(overbar) for equation (4). Please examine equations (4) and (8) and make changes where needed.

Thank you for your comment. As you correctly surmised, we intentionally included $\overline{V}_a$ to account for the background wind present in the real atmosphere. However, we agree that mathematically these terms cancel out, and for the definition of spectral width, the form excluding $\overline{V}_a$ (as in equations 5.102 and 6.17 in Fukao and Hamazu, 2014) is physically the most concise and accurate.

Please note that when excluding $\overline{V}_a$ in our equations 4 and 8, they are consistent with equation 6.17 in Fukao and Hamazu (2014). Equations 5.102 and 6.17 in Fukao and Hamazu (2014) are equivalent.

The description was redundant and potentially misleading, so we will revise the equation.

2. Lines 246-247. As written, I thought the phrase 'liquid phase' is referring to the Rayleigh scattering peak observed by the MU radar, but it appears to refer to the height region containing rain drops (aka, liquid phase). Please clarify that the 'liquid phase' refers to a height region containing liquid phase hydrometeors and not to the hydrometeors themselves.

Thank you for your feedback. We will change the description to "at all altitudes in the liquid phase."

3. Line 370. Is a word missing? How about, '...found that the XRAIN K_DP is sensitive..."

We apologize for the typo. We will correct it as you suggested.

4. Line 518. Attenuation in TAMX is discussed, but attenuation in NUX is not discussed. At these short ranges and larger particle regimes during stratiform rain, I do not believe that attenuation will impact the N0 estimate very much. But please confirm that assumption and please provide a sentence or two near equation (20) that discusses how attenuation in the measured NUX reflectivity is ignored or included in the estimate of N0.

As you pointed out, we have not taken into account the influence of NUX attenuation. The maximum observation height in this study is approximately 3 km from the NUX installed altitude. Even assuming a relatively high rain rate for stratiform precipitation (e.g., 12 mm h$^{-1}$ $\approx$ about 40 dBZ from $Z^* = 200R^{1.6}$ in Marshall et al., 1955), the specific attenuation at X-band is estimated to be around 0.2 dB km$^{-1}$ ($A = 0.011R^{1.15}$ of Table 6.1 in Fukao and Hamazu, 2014). Consequently, the total two-way attenuation over the 3 km path would be about 1.2 dB in the worst case. Although this value corresponds to an error of approximately 24% in linear reflectivity $Z^*$ and consequently in $N_0$, we consider this error acceptable for the purpose of this study.

We acknowledge that attenuation accumulates with height, which creates an artificial vertical gradient in the reflectivity profiles (i.e., underestimation increases with height). However, the magnitude of this artificial gradient in the worst case is estimated to be roughly 0.4 dB km$^{-1}$ (two-way). In contrast, the microphysical processes discussed in this study (e.g., evaporation, coalescence, and breakup) often cause larger vertical variations in radar reflectivity (Fig. 15a in the text). Furthermore, considering the NUX vertical resolution of 150 m, the difference of attenuation between adjacent range bins is estimated to be only about 0.06 dB. This slight increment is negligible compared to the measurement fluctuations. Therefore, although attenuation creates a slight artificial trend, it is not significant enough to mask the physical vertical evolution or alter the conclusions. We will add sentences near Equation 20 to clarify this assumption.

These are our responses to your comments and suggestions. Thank you in advance for your kind attention.

Yusuke GOTO (ISEE, Nagoya Univ., Japan)

**References**

Fukao, S. and Hamazu, K.: Radar for meteorological and atmospheric observations, Springer Tokyo, doi:10.1007/978-4-431-54334-3, 2014.

Marshall, J. S., Hitschfeld, W. and Gunn, K. L. S.: Advances in radar weather. *Advances in Geophysics*, Vol. 2, Academic Press, 1–56, doi:10.1016/S0065-2687(08)60310-6, 1955.